# Analytical methods for assessing retinal cell coupling using cut-loading

**William E. Myles** *, **Sally A. McFadden**

College of Engineering, Science and Environment, University of Newcastle, Callaghan, NSW, Australia

* will.myles@newcastle.edu.au

## Abstract

Electrical coupling between retinal neurons contributes to the functional complexity of visual circuits. "*Cut-loading*" methods allow simultaneous assessment of cell-coupling between multiple retinal cell-types, but existing analysis methods impede direct comparison with gold standard direct dye injection techniques. In the current study, we both improved an existing method and developed two new approaches to address observed limitations. Each method of analysis was applied to cut-loaded dark-adapted Guinea pig retinae (n = 29) to assess coupling strength in the axonless horizontal cell type ('a-type', aHCs). Method 1 was an improved version of the standard protocol and described the distance of dye-diffusion (space constant). Method 2 adjusted for the geometric path of dye-transfer through cut-loaded cells and extracted the rate of dye-transfer across gap-junctions in terms of the coupling coefficient ($k_j$). Method 3 measured the diffusion coefficient ($De$) perpendicular to the cut-axis. Dye transfer was measured after one of five diffusion times (1–20 mins), or with a coupling inhibitor, meclofenamic acid (MFA) (50–500µM after 20 mins diffusion). The standard protocol fits an exponential decay function to the fluorescence profile of a specified retina layer but includes non-specific background fluorescence. This was improved by measuring the fluorescence of individual cell soma and excluding from the fit non-horizontal cells located at the cut-edge ($p<0.001$) (Method 1). The space constant (Method 1) increased with diffusion time ($p<0.01$), whereas Methods 2 ($p = 0.54$) and 3 ($p = 0.63$) produced consistent results across all diffusion times. Adjusting distance by the mean cell-cell spacing within each tissue reduced the incidence of outliers across all three methods. Method 1 was less sensitive to detecting changes induced by MFA than Methods 2 ($p<0.01$) and 3 ($p<0.01$). Although the standard protocol was easily improved (Method 1), Methods 2 and 3 proved more sensitive and generalisable; allowing for detailed assessment of the tracer kinetics between different populations of gap-junction linked cell networks and direct comparison to dye-injection techniques.

## Introduction

Most of the cells of the retina are extensively linked by intercellular gap-junctions that allow the intercellular passage of ions and small molecules (typically up to 1000 Da) between pairs or

**Data Availability Statement:** Data are contained within the paper. Relevant analysis files are included in the supporting information.

**Funding:** This research was funded by the Hunter Medical Research Institute (Grant ID: G1801044)

and supported by the Lions Club, Newcastle. The funders had no role in study design, data collection and analysis, decision to publish, or preparation of the manuscript.

**Competing interests:** The authors have declared that no competing interests exist.

networks of coupled neurons. In the retina, cell coupling has been observed in all major cell types [1–4] either as homologous coupling amongst one cell-type or as heterologous coupling between multiple cell-types. These coupled networks facilitate the rapid, lateral spread of signals across the retina that complements slower synaptic transmission. Functionally, cell-coupling between retinal neurons modulates receptive field size, facilitates the transition between photopic and scotopic retinal states and contributes to signal averaging, synchrony and motion detection [5–7].

Gap-junctions are comprised of two linked hemichannels (connexons) spanning a 2-4nm intercellular gap, each formed by six connexin (Cx) proteins arranged radially surrounding a central pore [8, 9]. Twenty-one mammalian genes coding connexin proteins have been identified. Connexins are classified base on molecular weight, which ranges from 21 to 70kDa [10]. The cells of the retina express several connexin isoforms [6]. Retinal pigment epithelial cells express Cx43, cone and rod photoreceptors express Cx36 [11, 12], horizontal cells express Cx50 [13] or Cx57 [14], bipolar and amacrine cells express Cx36 [12] or Cx45 [15, 16], and retinal ganglion cells express Cx30.2 [17], Cx36 [18], or Cx45 [19].

The conductance of gap-junctions may be transiently regulated by the phosphorylation of serine residues in the component connexin proteins [20–22]. In the retina, this occurs in response to the release of light-mediated neuromodulators such as dopamine [23, 24] and nitric oxide [25, 26], facilitating in the switching between light and dark processing pathways. For example, heterotypic Cx36/45 gap-junctions between AII amacrine cells and ON-cone bipolar cells are disinhibited in scotopic illumination, allowing for rod signals to utilise the downstream cone pathway [2].

Given the fundamental importance of coupled networks in determining the fate of signal processing in the retina, understanding their role relies on measurement of coupling after various experimental manipulations. Methods for assessing cell-cell coupling include simultaneous electrical recordings from cell pairs [27–29], intracellular microinjection of molecular tracers [30, 31], and cut-loading [32, 33]. Patch-clamping and microinjection techniques provide an exceptional level of detail, however the techniques themselves are highly technical, requiring the prior identification and recording from individual cells or cell pairs. Cut-loading is a simple alternative whereby the retina is cut with a scalpel blade dipped in molecular tracer and the relative diffusion through coupled cells is measured at a fixed time point.

The most common technique for analysing cut-loaded tissues is to use the space constant from an exponential decay function fitted to the fluorescent intensity of the tissue measured per unit distance from the cut location [32, 34, 35]. This technique has several limitations in that: 1) it does not account for variation in cell density between tissues or sampling location, 2) the equation is fitted to data describing the mean fluorescence of an area containing both cells and non-specific background fluorescence, 3) the technique is limited to studying cell populations isolated in a particular layer of the retina e.g., photoreceptors or horizontal cells and 4) space constant values obtained via this method cannot be directly compared to those obtained via other techniques.

Other techniques such as using compartmental analysis to determine coupling coefficients $k_j$ between coupled cells [36], or calculating the effective diffusion coefficient of molecular tracers through coupled cell networks [37] have been employed to measure dye-diffusion from single cells loaded via microinjection techniques. These techniques have also been applied to microinjection experiments using the intracellular tracer Neurobiotin™, in which only single time-point data is available [38], however, few studies have integrated this analytical approach as part of the cut-loading protocol [33, 39].

In the present study, we adapted three analytical techniques for analysing cell-coupling in cut-loaded mammalian retinae and compared their effectiveness. Method 1 was based on the

standard exponential decay analysis protocol, however we employed several simple procedural modifications to help overcome its inherent limitations: 1) the fluorescence of each cell soma was fitted, rather than the mean fluorescence profile of the image as a whole 2) highly fluorescent un-coupled cells located at the cut were excluded from the fit 3) each image was adjusted by the mean cell-cell spacing to account for variation in signal spread due to changes in cell density. These methodological adjustments were also included for the remaining two methods. Method 2 utilised a model based on Zimmerman and Rose's method for calculating the coupling coefficient $k_j$ [36], however, our method also adjusted for the geometric arrangement of retinal cells and the path of molecular tracer based on a line of cut-loaded cells feeding a broad network. Method 3 applied Fick's second law of diffusion to calculate the effective diffusion coefficient $De$ in one-dimension (along the axis perpendicular to the cut) through a network of coupled cells. We compared these three techniques in retinas across five dye incubation times and in retinas incubated for a set incubation time with increasing concentration of the gap-junction inhibitor meclofenamic acid (MFA). We report that the standard analysis protocol may be improved via some simple procedural modifications, however, Methods 2 and 3 were ultimately more versatile and provided greater overall sensitivity, with the advantage of allowing direct comparison of diffusion kinetics with data from dye-injection methods.

## Methods

### Animals

Tri-coloured domestic Guinea pigs (*Cavia porcellus*, n = 29) were reared in plastic boxes with stainless-wire raised tops, illuminated overhead by white light emitting diodes diffused through a 3mm thick Perspex screen located 200 mm above the box lid. Lights operated on a 12-hour light-dark cycle at a constant illuminance of 700lx, measured at the animal's eye level within the housing box. Room temperature was maintained at 21(±2)°C and food and water was provided *ad libitum*. All experiments and experimental procedures were approved by the Animal Care and Ethics Committee of the University of Newcastle and were conducted in accordance with the Australian code for the care and use of animals for scientific purposes.

### Experimental design

Two experiments were undertaken on retinae freshly extracted from dark adapted guinea pigs between 14 and 28 days of age (Table 1). Experiment 1 used 17 retinae that were cut-loaded with molecular tracer for different incubation times, and the imaged tissues were analysed using three different methods to yield the coupling space constant, the coupling rate constant, or the diffusion coefficient. In Experiment 2, retinae were cut-loaded with molecular tracer and incubated for 20 minutes. Cell coupling was inhibited with increasing concentrations of the general gap-junction inhibitor meclofenamic acid and imaged tissues were analysed using the same three methods as in Experiment 1. In both experiments, the axonless horizontal cell type ('a-type', aHC) were selected as a model system for analyses. In Experiment 3, the generalisability to a different coupled network was studied in retinae from Experiment 2, by co-labelling for neuronal nitric oxide synthase (nNOS).

### Procedures

**Light adaptation and euthanasia.** Guinea pigs were adapted to complete darkness for one hour prior to euthanasia. Animals were anaesthetised using gaseous isoflurane (5% in 1.5 L/min $O_2$) and euthanised via intracardial injection of Pentobarbitone Sodium (Lethabarb®, Virbac Australia Pty Ltd, NSW, Australia). A dim red head torch (light emitting diode (LED)

**Table 1. Experimental design showing the primary variables manipulated.** The *Experimental Light Exposure* refers to the light level during cut-loading at a wavelength of 850nm and was approximately equivalent to total darkness.

| | Aim | Incubation Time (mins) | No. of retinae (No. of cuts) | MFA Concentration (μM) | Experimental Light Exposure (log $R^*.Rod^{-1}.s^{-1}$) |
|---|---|---|---|---|---|
| **Experiment 1** | Test analysis methods across different incubation times (Horizontal Cells) | 1 | 2 (5) | N/A | -4.5 |
| | | 3 | 4 (10) | | |
| | | 5 | 4 (12) | | |
| | | 10 | 4 (10) | | |
| | | 20 | 3 (7) | | |
| **Experiment 2** | Test analysis methods for variable coupling at a fixed time-point (Horizontal Cells) | 20 | 2 (4) | 50 | -4.5 |
| | | | 2 (4) | 100 | |
| | | | 2 (5) | 150 | |
| | | | 2 (5) | 200 | |
| | | | 2 (4) | 250 | |
| | | | 2 (4) | 500 | |
| **Experiment 3** | Reanalysing tissues from Exp. 2, for non-horizontal cells | Same as Exp. 2 | | | |

spectral peak at 635nm, max 20uW.cm$^{-2}$ at 20cm away from tissue, luminance at tissue was 0.5 log $R^*.Rod^{-1}.s^{-1}$) was used during euthanasia and dissection in an otherwise dark room. During cut-loading, infrared LEDs (spectral peak 850nm, -4.5 log $R^*.Rod^{-1}.s^{-1}$) and night vision goggles (Sionyx Aurora Pro) were used to maintain dark-adaptation of the retina during experimentation.

**Cut-loading procedure.** The eye was rapidly enucleated and submerged in Ames solution (catalogue: A1420-10X1L, Sigma-Aldrich, MO, USA) at room-temperature (18 to 21˚C). Whilst submerged, the cornea and limbus were removed with a circular cut extending along the posterior pars plana (Fig 1). A small incision was then made along the long ciliary artery to mark the nasal axis using a No.11 scalpel blade. The crystalline lens and vitreous humour were gently removed using No.5 forceps. The remaining eye cup containing the retina, choroid and sclera were transferred into a well plate containing 10mL of Ames solution (36˚C) bubbled with carbonox (95% $O_2$ 5% $CO_2$).

The retina was separated from the underlying choroid using a blunt dental spatula. The optic nerve joining the retina and sclera was then cut using 3mm curved scissors and the retina orientated with photoreceptor side down was mounted onto 0.22μm pore size Millipore membrane filter paper (catalogue: GSWP04700, Merck & Co. NJ, USA). Tissues were acclimatised for 15 minutes in complete darkness in the bubbled Ames solution. In Experiment 2, the required dilution of meclofenamic acid (MFA) (catalogue: M4531, Merck & Co. NJ, USA) made from stock solution of 100mg/mL MFA sodium salt dissolved in 100% ethanol was added directly to the acclimatisation Ames bath solution and all bath solutions up until fixation.

The retinas were briefly removed from solution and cut along the superior, temporal, and inferior axes with a size 11 scalpel blade that prior to each cut was dipped in 3% the biotin derivative, N-(2-aminoethyl) biotinamide hydrochloride (Neurobiotin™ Tracer, catalogue: SP-1120, Vector Laboratories, CA, USA) diluted in Ames solution. 5% wt/v Rhodamine dextran B 10,000 MW (catalogue: D1824, ThermoFisher Scientific, MA, USA) was added to the cut solution for one retina from each of the 1, 3, 5 and 10 minute incubation groups in experiment 1. The tissue was returned to the Ames solution and the Neurobiotin™ dye allowed to diffuse through the cell network by incubating for either 1, 3, 5, 10 or 20 minutes in Experiment 1, or 20 minutes in Experiment 2 (see Table 1). At the end of each Experiment, tissues were

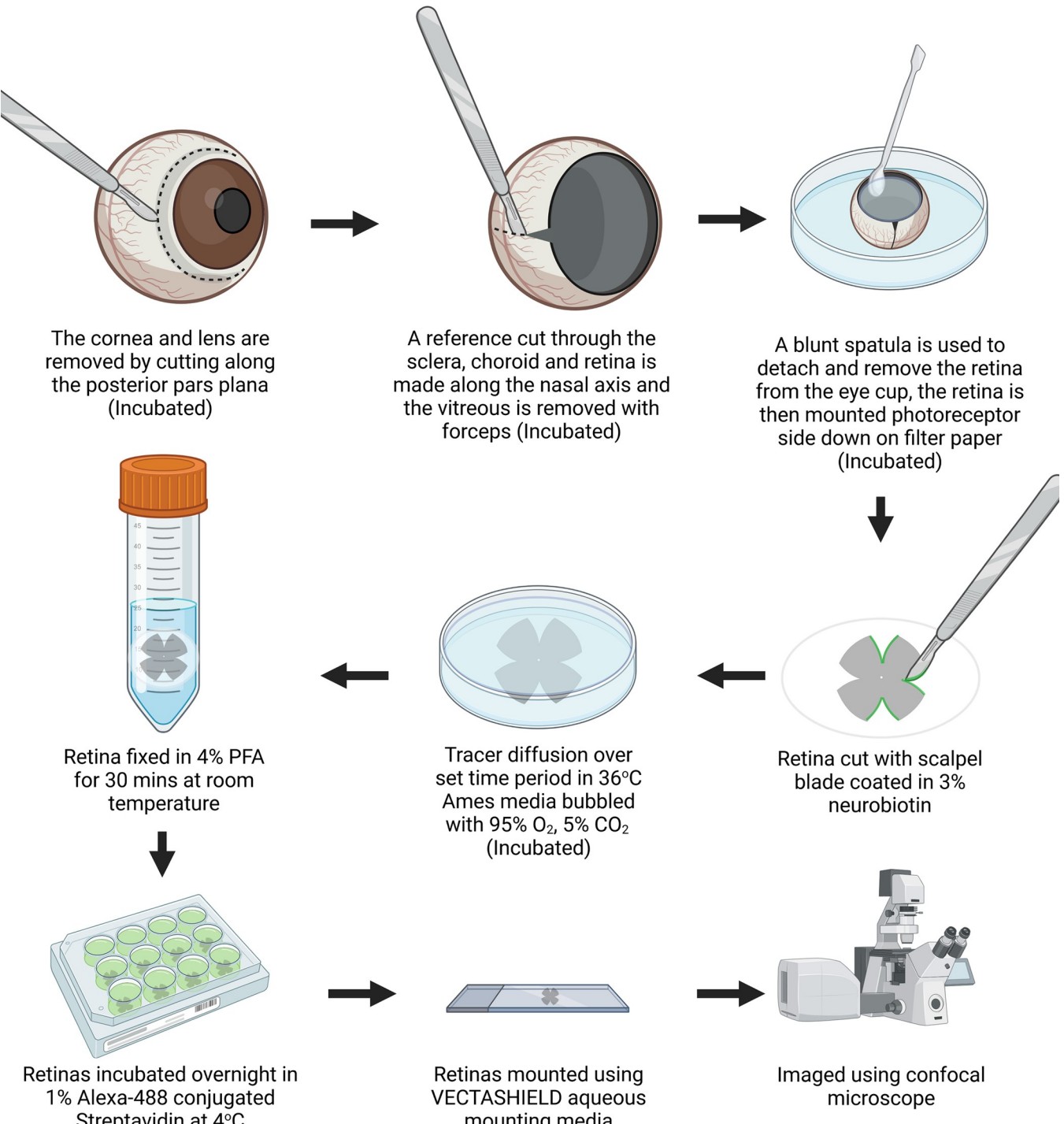

**Fig 1. Flow chart illustrating the key steps in the cut-loading procedure.** Figure made using the online application BioRender.com.

removed from the bubbled bath solution and fixed in 4% paraformaldehyde (4% wt/v, diluted in 0.1M phosphate buffer) at room temperature for 30 minutes. Retinas were washed in 1 x PBS (30 mins) and reacted with Alexa-Fluor 488 conjugated streptavidin (catalogue: S11223, ThermoFisher Scientific, MA, USA, diluted 1:100 in 0.5% Triton-x in PBS) overnight at 4°C.

**Table 2. Antibody reagent list.** Both primary and secondary antibodies were diluted in 1% NDS and 0.5% Triton-x in PBS. RT, room temperature.

| Antibody | Catalogue | Host Species | Company | Dilution | Time (Temperature) |
|---|---|---|---|---|---|
| Neuronal Nitric Oxide Synthase (nNOS) | N7280 | Rabbit | Sigma-Aldrich | 1:1000 | Overnight (4°C) |
| Calbindin D-28K | AB1778 | Rabbit | Sigma-Aldrich | 1:400 | Overnight (4°C) |
| Donkey anti-Rabbit (CY3) | 715-165-150 | Donkey | Jackson ImmunoResearch Laboratories, Inc. | 1:400 | 1 hour (RT) |

Retinas were washed in 1 x PBS and mounted, photoreceptor side down, onto microscope slides using anti-fade Vectashield aqueous mounting medium (Vector Laboratories, CA, USA, catalogue: H-1000-10).

**Immunofluorescence.** Retinae from Experiments 1 and 2 were removed from glass slides and placed in PBS, incubated in Triton-x for 30 minutes (1% Triton-x in PBS) and blocked using normal Donkey serum (10% NDS in 0.5% Triton-x, PBS) for one hour. Retinae from Experiments 1 and 2 were then counter labelled with antibodies for Calbindin and Neuronal Nitric Oxide Synthase (nNOS) respectively (See Table 2 for details). Retinae were washed in PBS (3 x 10 mins) and incubated in secondary antibody solution (Table 2). Retinas were washed in 1 x PBS and mounted, photoreceptor side down, onto microscope slides using anti-fade Vectashield aqueous mounting medium.

**Image collection and preparation.** Images were collected 3mm from the optic nerve head along the superior, temporal and inferior cuts using an Olympus IX91 scanning laser confocal microscope. Each image comprised of 50 optical slices (1μm apart), taken between the outer plexiform later and the retinal ganglion cell layer. Confocal settings were kept consistent across all image acquisition. Composite images used for analysis were created from 10 slices selected from the z-stack using the sum-slices z-project function in Fiji (open-source distribution based on ImageJ2 released by National Institutes of Health) [40]. For aHCs these 10 slices spanned from the boundary of the outer plexiform layer and the inner nuclear layer.

**Light intensity calculations.** Reported values of spectral irradiance ($E(\lambda)$, $\mu W.cm^{-2}.nm^{-1}$) of the home-box during rearing, the red LED head torch used during euthanasia and dissection and the infrared LED lighting used during cut-loading experimentation were measured using an Ocean Optics spectrophotometer (USB-4000, Ocean Optics). Illuminance (lx) was calculated based on the Guinea pig photopic spectral sensitivity curve [41]. Photometric units were converted to total effective $photons.cm^{-2}.s^{-1}$ based on the rod spectral sensitivity curve as determined using Govardovskii nomograms [42] with $\lambda_{max}$ = 496 nm (Jacobs & Deegan, 1994). This was then converted into $photoisomerisations.rod^{-1}.s^{-1}$ ($R^*.rod^{-1}.s^{-1}$) assuming a 1.0 $\mu m^2$ effective collection area for each rod [28]. At low light levels, irradiance ($\mu W.cm^{-2}$) was measured using a Newport optical power meter centred at the spectral peak (main unit: 2936-R, sensor: 818-ST2-UV/DB, Newport).

**Modelling Neurobiotin$^{TM}$ dye-transfer.** As the true cellular concentration of dye was not known, the relative mean fluorescence of each cell was used to gauge the relative concentration of tracer in cells.

*Standard protocol*. The mean fluorescence of the image spanning perpendicular from the cut location was measured using the plot profile function in Fiji [40]. The fluorescence profile was then fitted with the exponential decay curve below:

$$C = C_o e^{\left(\frac{-x}{\lambda}\right)} \tag{1}$$

Where $C$ is the concentration of the cell at distance x from the cut ($\mu m$), $C_0$ is the maximum concentration and $\lambda$ is the space constant ($\mu m$).

*Modified protocol (Method 1)*. The mean fluorescence and position of each cell soma was measured in Fiji using the oval tool. The background fluorescence of the retinal tissue was

measured for each composite image in a region approximately 1500μm from the cut which contained no fluorescing cells and was subtracted from absolute fluorescence measurements. All measurements were then normalised to the maximum fluorescence measured at the cut location. The distance between adjacent horizontal cells (Experiments 1 and 2) and amacrine cells (Experiment 3) were measured from soma centre to soma centre. Forty measurements for each cell-type per retinal image were used to obtain an average intercellular distance for each image. Data were analysed as both cell fluorescence per absolute distance (μm) from the cut site by fitting cell fluorescence data with Eq (1), and as cell fluorescence per cell-separation from the cut site, by dividing the absolute distance of the cell from the cut by the mean soma-soma distance. In the latter instance, the units of x become cell-separations and the units of λ become cells.

*Method 2.* Zimmerman and Rose [36] described the diffusion kinetics of molecular tracers through a chain of 5–7 coupled giant cells of the Chironomus salivary gland using compartment model analysis. For a series of three coupled cells ($C_0$, $C_1$, $C_2$) the movement of tracer into ($q_r$) and out of the cell ($q_s$) $C_1$ can be approximated using the following equations:

$$q_R = -k_j(C_1 - C_0) \tag{2}$$

$$q_s = -k_j(C_2 - C_1) \tag{3}$$

Where $k_j$ is a rate constant of units (distance$^2$.time$^{-1}$). The net movement of tracer through cell $C_1$ can then be approximated by the combination of these two equations to yield:

$$q_R - q_S = -k_j(C_1 - C_0 - C_2 + C_1) = k_j(C_0 + C_2 - 2C_1) \tag{4}$$

The geometric arrangement of HCs differs to that of giant cells used in Zimmerman and Rose's original paper. The arrangement of HCs can be modelled using a triangular lattice, with each cell connecting to the six neighbouring cells (Fig 2). Therefore, in cut-loading a line of

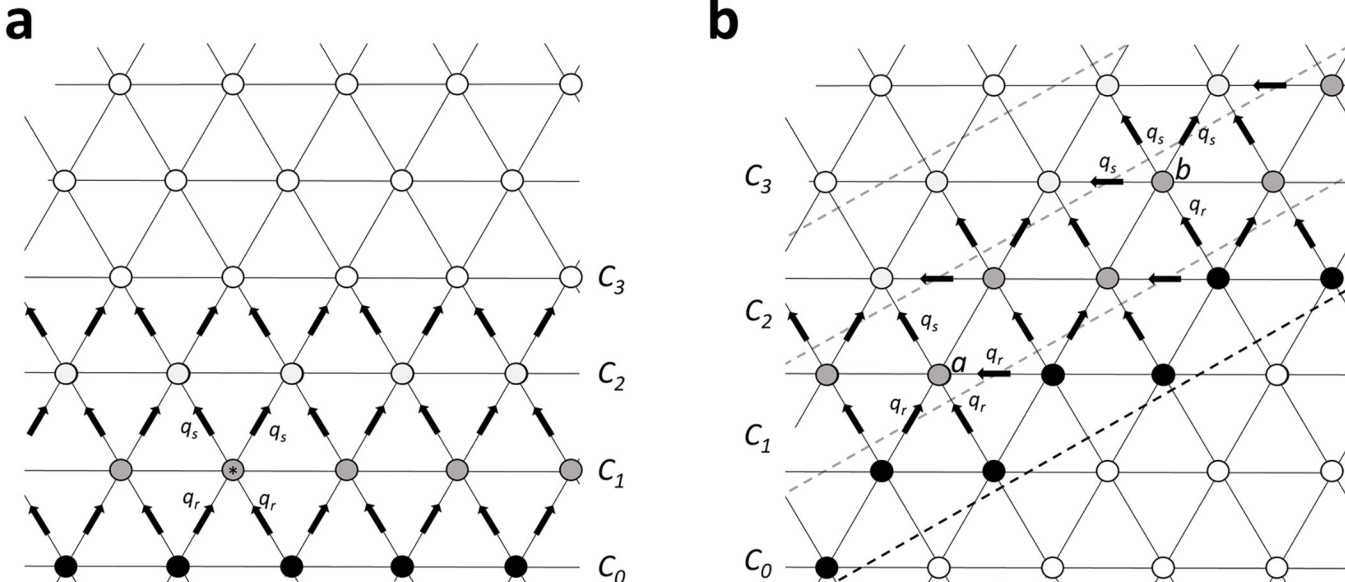

**Fig 2. Geometric model of dye-transfer through coupled horizontal cells. (a)** Cells located in $C_1$ (*) receive dye ($q_r$) from two cells in $C_0$ and individually feed two cells ($q_s$) in $C_2$. **(b)** Cells located in $C_1$ either receive input from three cells in $C_0$ and feed one in $C_2$ (cell *a*), or they receive input from one cell in $C_0$ and feed three cells in $C_2$ (cell *b*).

cells will be initially filled with tracer. If we assume no net tracer transfer occurs between cells of equal internal dye concentration, then the number of cells $C_0$ would feed into depends on the angle of the cut with respect to the lattice (Fig 2). In situation one, each cell at $C_1$ would receive tracer from two cells at $C_0$ and would feed into two cells at $C_2$ (Fig 2A). In situation two, the cells at $C_1$ would either receive tracer from one or three cells at $C_0$ and would feed into either one or three cells at $C_2$ (Fig 2B). As these situations alternate in the second example, each layer receives and exports tracer to an average of two cells.

This was incorporated into Eq (4) to yield:

$$2q_R - 2q_S = -k_j(2C_1 - 2C_0 - 2C_2 + 2C_1) = 2k_j(C_0 + C_2 - 2C_1) \tag{5}$$

Which yields the series:

$$\frac{dC_1}{dx} = 2k_j(C_2 - C_1) - k_s(C_1 V_1) \tag{6}$$

$$\frac{dC_n}{dx} = 2k_j(C_{n+1} + C_{n-1} - 2C_n) - k_s(C_n V_n) \text{ etc.}$$

Where $k_j$ is a rate constant describing dye transfer between coupled cells within a network (cells$^2$/s), $k_s$ is sequestration or loss of dye as it passes through the tissue (cells$^2$/s) and $V$ is the relative volume of the cell (set to 1 as all HC assumed to have equal volume). These equations were solved in MATLAB (mathworks) by first fitting a 2-parameter Gaussian curve to the original data to calculate the mean relative fluorescence at discrete intervals ($C_n$). A solution for $k_j$ was obtained by fitting the concentration series to the ode45 solver, which solved the above equation series (expanded to n = 1:45) at a defined time point based on the 4th and 5th Runge-Kutta method. The MATLAB code used in the present study for cut-loading analysis is included in S1 File.

When $k_j$ was calculated in terms of absolute distance (cm$^2$.s$^{-1}$), the geometric path of molecular tracer was not accounted for and the non-normalised cell-cut distances were used for calculation via Zimmerman and Rose's [36] original equation written below:

$$\frac{dC_1}{dx} = k_j(C_2 - C_1) - k_s(C_1 V_1) \tag{7}$$

$$\frac{dC_n}{dx} = k_j(C_{n+1} + C_{n-1} - 2C_n) - k_s(C_n V_n) \text{ etc}$$

*Method 3*. The diffusion of dye through a homologously coupled cell network (such as horizontal cells) following cut-loading can be described as the diffusion of a substance along one axis using Fick's second law of diffusion [43]:

$$\frac{\partial C}{\partial t} = D\frac{\partial^2 C}{\partial x^2} \tag{8}$$

Where $C$ is concentration of the diffusing substance, $t$ is time in seconds, $D$ is the diffusion coefficient (cm$^2$.s$^{-1}$) and $x$ is distance (cm).

One method for modelling dye-diffusion during cut-loading is to consider that at $t = 0$, all molecular tracer is located within the region $-l < x < +l$ describing the boundary of the initial cut made through the retina. Dye transfer then occurs in one-dimension through the coupled cells at the cut boundary, which can be treated as the sum of an infinite number of line sources

with diffusion occurring along the x-axis and modelled using the following equation:

$$C = \frac{C_0}{2} \left\{ erf\left(\frac{l - |x|}{2\sqrt{Dt}}\right) + erf\left(\frac{l + |x|}{2\sqrt{Dt}}\right) \right\} \qquad (9)$$

Where:

$$erf\,(z) = \frac{2}{\sqrt{\pi}} \int_0^z e^{-t^2}\, d\,t \qquad (10)$$

And $C_0$ is the initial concentration, with initial boundary conditions at t = 0:

$$C = C_0,\ x < l \text{ and } C = 0,\ x > l$$

Fig 3 shows an example solution for Eq (9). with $l = 0.04$ cm, $D = 1.6$ x $10^{-6}$ cm$^2$.s$^{-1}$ and $C_0 = 6.15$.

**Statistical analysis.** Data from the five incubation conditions in Experiment 1 were analysed using a 1x5 way ANOVA in SPSS (IBM Statistics 25). In cases where data failed the assumption of homogeneity of variance (Levene's test, $p < 0.05$) a Brown-Forsythe ANOVA was used instead. Independent samples t-tests were used when comparing fits (Experiment 1) and when comparing coupling rates between cell types (Experiment 3). In Experiment 2, the reduction in cell-coupling due to MFA (100 μM– 500 μM) was expressed as a percentage reduction from the low-dose (50 μM MFA) experimental condition. Paired samples t-tests were then used to compare the normalised effective response calculated by the three analysis

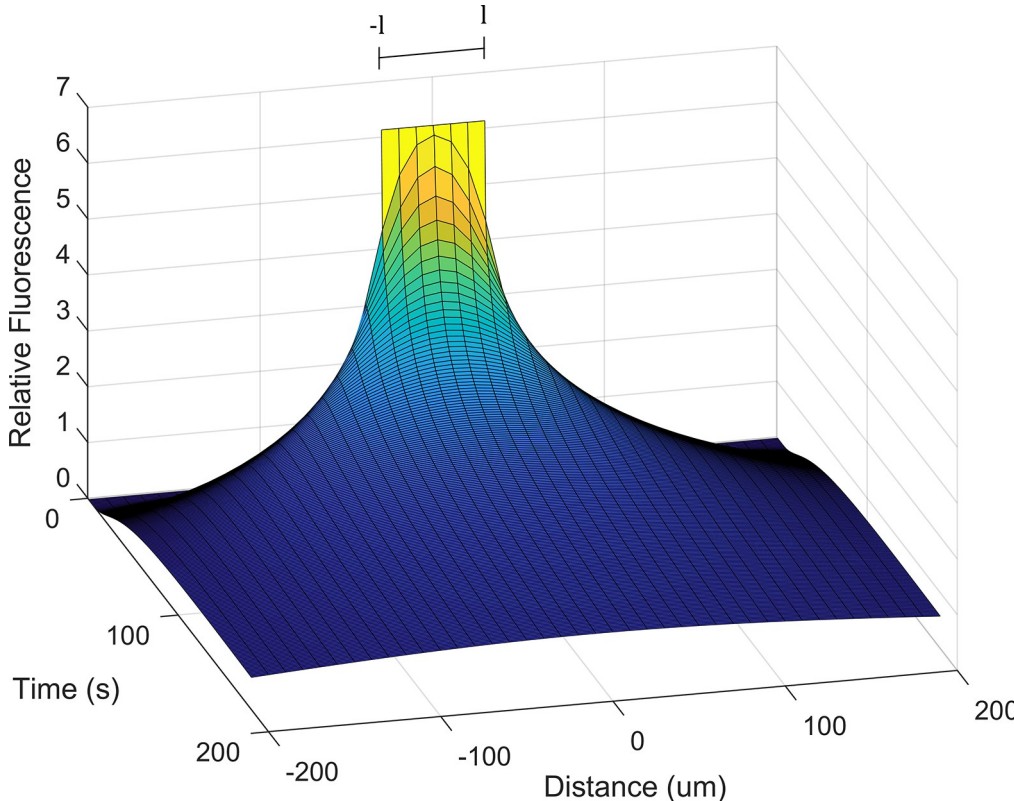

**Fig 3. The diffusion of molecular tracer from the cut location, modelled according to Eq (9).** At $t = 0$, all molecular tracer is located within the region $-l < x < l$ surrounding the x-axis. Concentration of Neurobiotin$^{\text{TM}}$ is represented as relative fluorescent intensity along the z-axis.

methods (Experiment 2) at MFA concentrations 150 μM and 250 μM, representing a partial and full inhibition respectively. Wilcoxon signed-rank test was used when data was not normally distributed. All tests were conducted as two-tailed tests. Data from Experiment 2 was fitted with the dose-response Hill curve [44]:

$$\frac{I}{I_{max}} = \frac{1}{1 + \left(\frac{IC_{50}}{[A]}\right)^{-n}}$$

(11)

Where I is the magnitude of the change in coupling, $I_{max}$ is the maximum change in coupling, [A] is the concentration of meclofenamic acid, $n$ is the Hill coefficient and $IC_{50}$ is the concentration at which 50% of the maximum response is observed. Fitting was performed in Matlab (Mathworks) using a modified script by Ritchie Smith [45]. Power analyses were performed in G*Power (Version 3.1, Universität Kiel and Universität Dusseldorf, Germany) for an independent samples t-test (2-tailed).

## Results

### Modification of the standard analysis protocol

An example cut-loaded retinal image of Neurobiotin™ filled aHCs spanning 1.5 mm from the cut is presented in Fig 4A, the associated fluorescence data in Fig 4B, and the same data translated onto a log scale in Fig 4C. The exponential decay function from the standard cut-loading analysis protocol, Eq (1), produced a poorer fit to the data in regions closest to and furthest away from the cut (red dashed line in Fig 4C) but overall was well fitted (mean $R^2$ of fit ± S.D = 0.89 ± 0.05, $n$ = 44 cuts). This layer contained both highly coupled aHCs as well as the less coupled axon-bearing 'b-type' HCs (bHCs). These subtypes were easily distinguishable based on their distinct cellular morphology (Fig 4C). Both subtypes co-labelled with calbindin (Fig 4D), however, Neurobiotin™ was only observable in bHCs within 200μM from the cut, whereas Neurobiotin™ dye was observed in aHCs up to 1500μM from the cut (Fig 4A and 4E).

An alternative method to the standard protocol (Method 1) was based on the mean fluorescence for each aHC soma (Fig 4F, blue dashed line fit to grey closed markers) and had the advantage of excluding background fluorescence that was not relevant to the cells of interest and excluding fluorescence from dye-loaded bHCs, observed immediately adjacent to the cut (Fig 4F, purple dashed line fit to purple markers). Therefore, this approach gave a more accurate estimate of the dye transfer within the cell network with increasing distance from the cut. The median decay of the exponential fit for aHCs was significantly less rapid, corresponding to a larger calculated space constant than that produced by fitting the same equation to the total mean image fluorescence (270.3 vs. 212.7μm, Fig 4C, $Z$ = 3.74, $p$ < 0.001). Importantly, data can also be meaningfully normalised for mean soma-soma distance between adjacent aHCs (Fig 4D) to adjust for variations in cell density between retinae and sampled areas. This alternative technique was employed in Experiments 1 and 2 as Method 1.

### Experiment 1. Time-dependant dye transfer through coupled a-type horizontal cells

The extent to which Neurobiotin™ travelled through coupled aHCs increased with incubation duration (Fig 5A–5E).

The fluorescence of cells situated immediately adjacent to the cut was less than that observed deeper into the tissue for all incubation conditions (Fig 5H), resulting in non-linear dye-transfer characteristics (compare open and closed markers in Fig 6A–6C). The aHCs

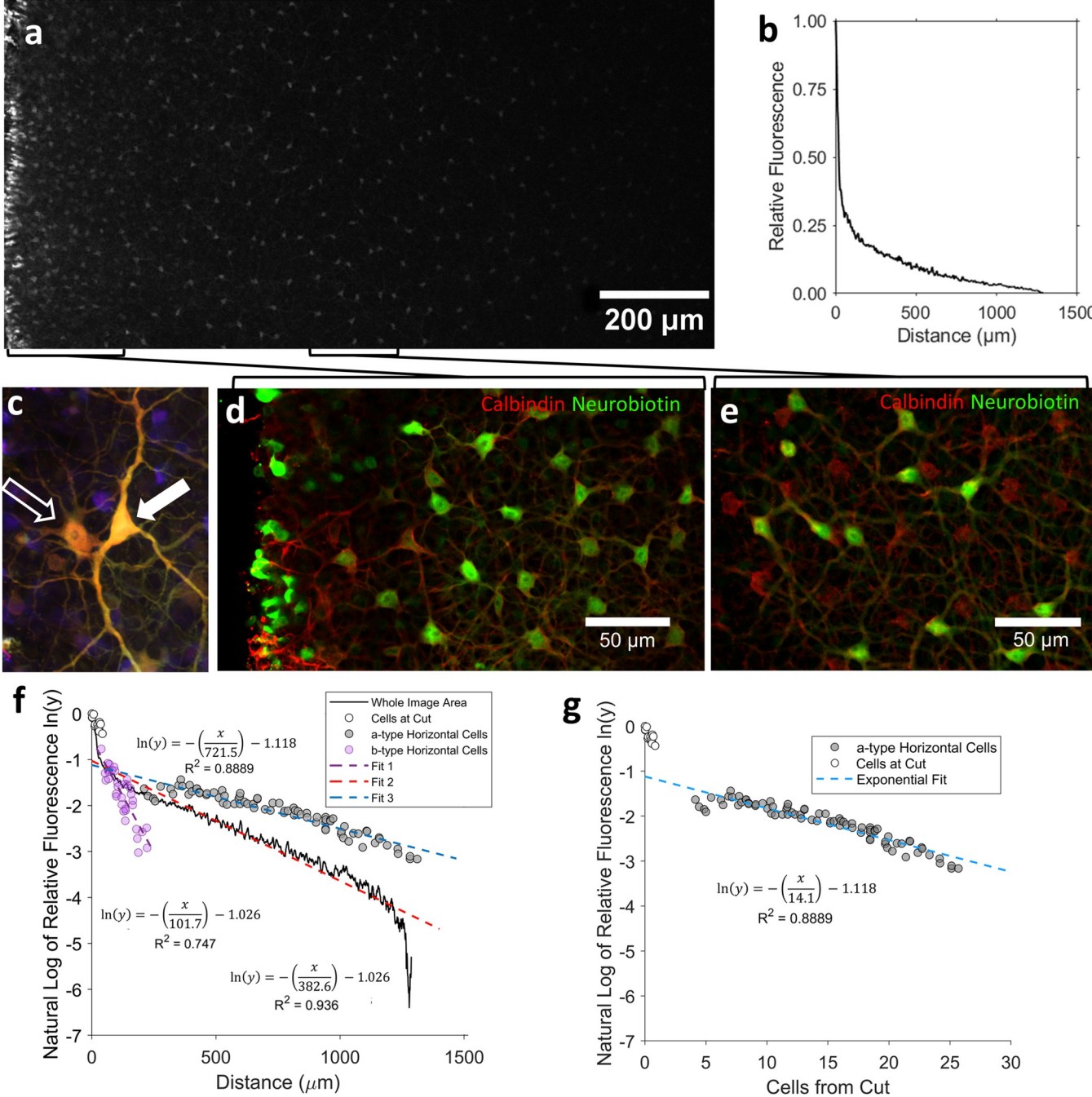

**Fig 4.** **(a)** Example image taken in outer portion of the inner nuclear layer of a Guinea pig retina cut-loaded with Neurobiotin[TM]. **(b)** The fluorescent intensity of the whole image measured along the horizontal axis extended away from the cut. **(c)** Close up of morphologically distinct horizontal cell subtypes: axon-bearing ('B-type', open arrow) and axon-less ('A-type', closed arrow) HCs. Image created from cut-loaded retina using the temporal colour-code function in Fiji for optical slices spanning the inner nuclear layer (see methods). **(d)** Cut-loaded aHCs and bHCs located adjacent to the cut, loaded with Neurobiotin[TM] (green) and co-labelled with Calbindin (red). **(e)** Cut-loaded aHCs located 500 µm from the cut, loaded with Neurobiotin[TM] (green) and co-labelled with Calbindin (red). **(f)** Semi-log plot of fluorescent intensity data from **(b)** (dashed red line) and fluorescent intensity of individual aHCs (grey markers, dashed blue line) and bHCs (purple markers, dashed purple line)), each fitted with an exponential decay function, Eq (1). **(g)** Fluorescent intensity of individual aHCs adjusted for the mean soma-soma cell distance.

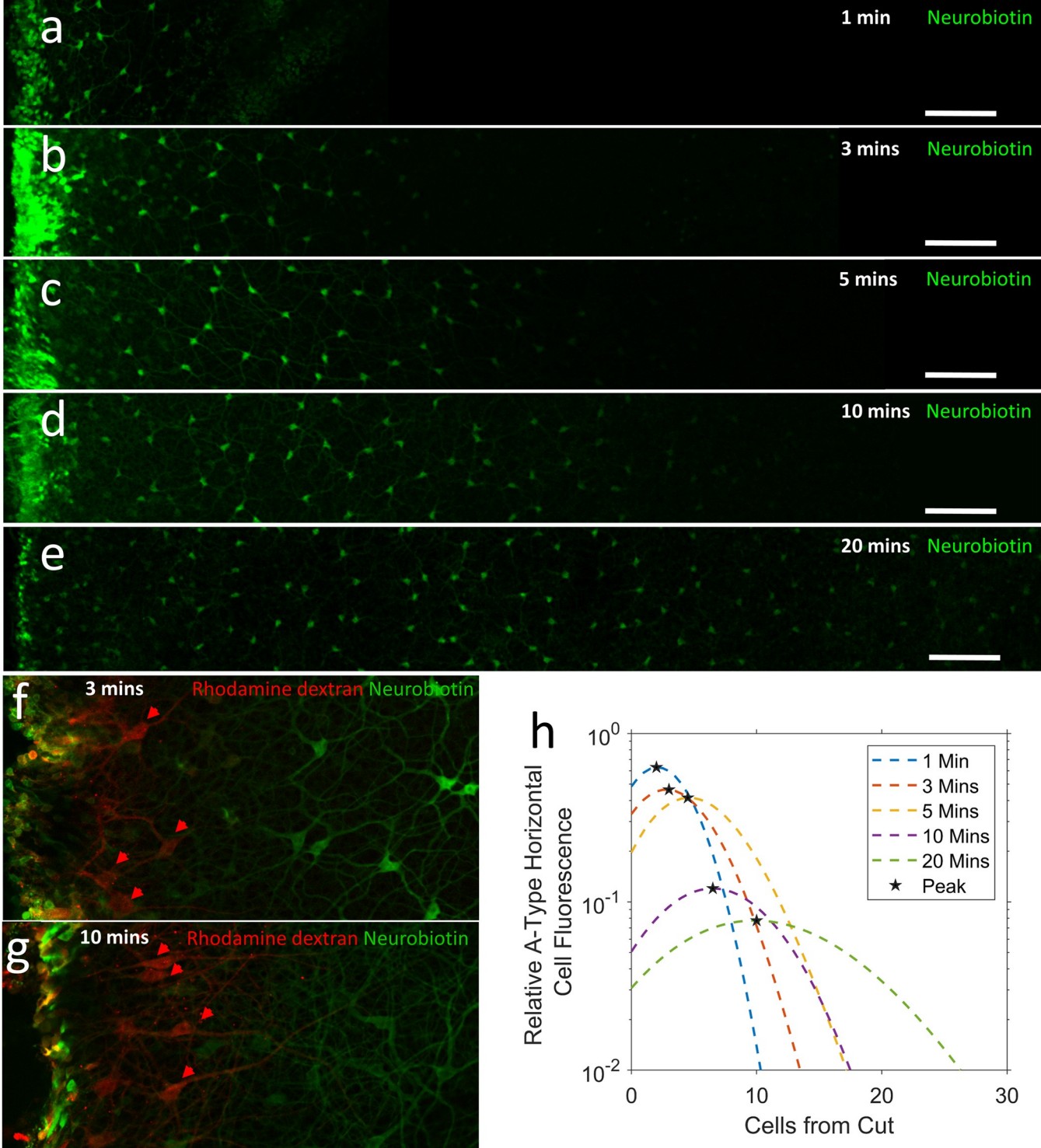

**Fig 5.** **(a-e)** The extent of dye-transfer through coupled a-type horizontal cells (aHCs) after different incubation times between 1 and 20 min. Images represent the average of 5, 1μm thick optical slices through the outer inner nuclear layer created using the z-project function in Fiji (see methods). The cut-edge is located on the left-hand side of each image. **(f, g)** aHCs cut-loaded with Neurobiotin (green) and the gap-junction impermeable dye Rhodamine dextran (red, aHCs identified with red arrows) after 3 and 10 min incubation respectively. **(h)** Relative mean cell fluorescence of aHCs per cell from the cut. Curves produced by fitting 3-parameter polynomial fits to mean aHC data for each incubation time, fluorescence at the cut was excluded when creating the fit. The peak cell fluorescence for each curve is indicated by black stars. Scale bars signify 100μm.

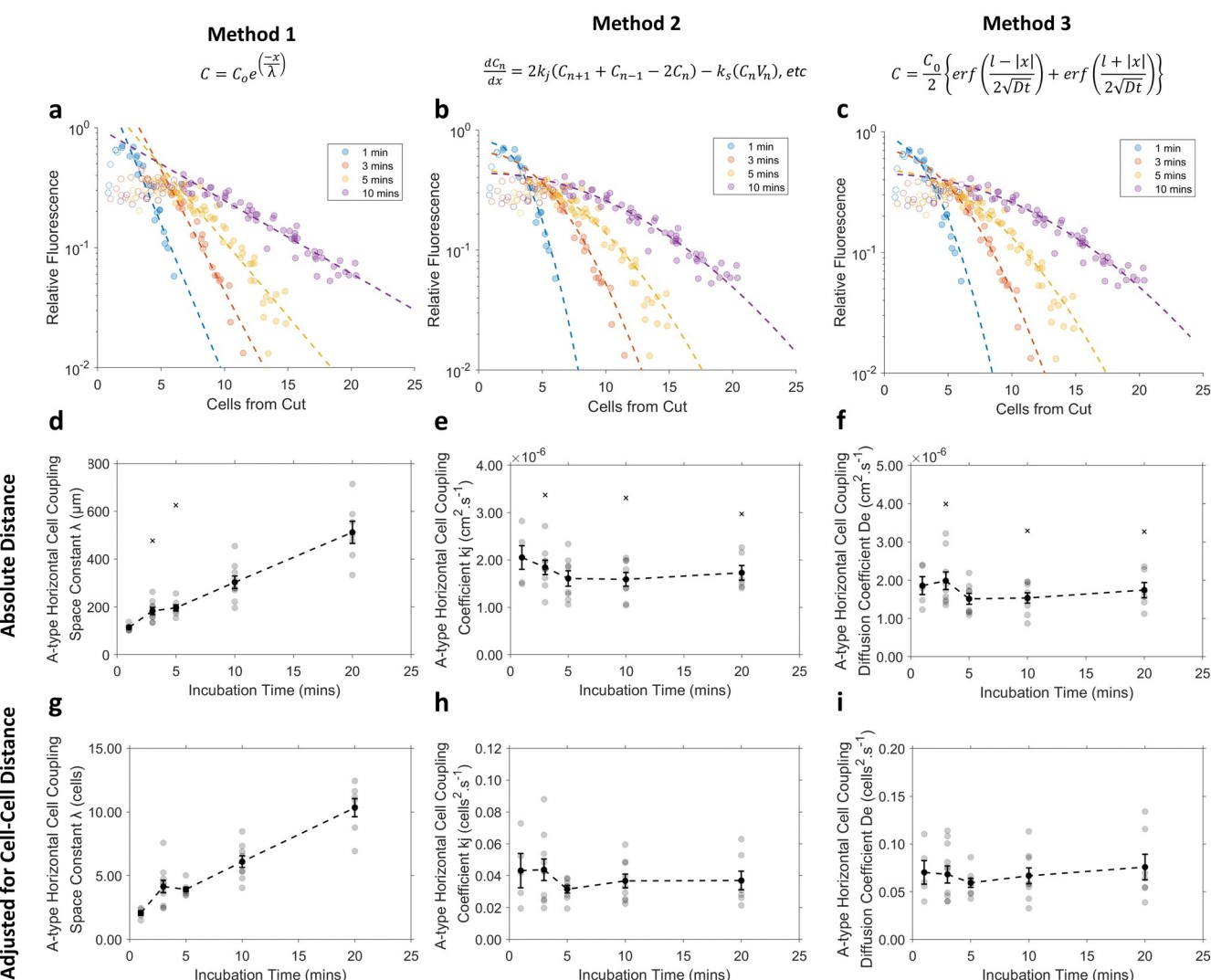

**Fig 6. Comparison of three analytical techniques for assessing the extent of a-type horizontal (aHCs) cell coupling in cut-loaded Guinea pig retinas with increasing incubation times.** Column 1 (**a, d, g**) contains cell fluorescence data analysed using Method 1. Column 2 (**b, e, h**) shows the same data analysed using Method 2 and column 3 (**c, f, i**) shows the same data analysed using Method 3. The first row (**a, b, c**) contains example fits for cell fluorescence data collected per each cell separation following four incubation times, fitted with (**a**) Eq (1), (**b**) Eq (6) and (**c**) Eq (9) corresponding to Methods 1, 2 and 3 respectively. Open markers indicate cells adjacent to the cut with reduced fluorescence. Row 2 (**d, e, f**) contains the mean ± SE (black marker, black error bars) and individual measurements (grey markers) for each measure of cell-coupling obtained via: (**d**) Method 1 (**e**) Method 2 and (**f**) Method 3. Row 3 (**g, h, i**) contains the mean ± SE (black marker, black error bars) and individual measurements (grey markers) for each measure of cell-coupling normalised to the mean cell-cell spacing between adjacent aHCs, obtained via: (**g**) Method 1 (**h**) Method 2 and (**i**) Method 3. Crosses indicate outliers defined as data points greater than 1.5 standard deviations from the mean and are not included in calculated mean.

initially loaded during cut-loading are identified by the presence of the impermeable dye Rhodamine dextran (Fig 5F and 5G red fluorescence, aHCs identified by red arrows). Interestingly, these aHCs display little or no fluorescence from Neurobiotin following 3 and 10 minutes tissue incubation. Furthermore, the distance from the cut where reduced aHC fluorescence occurred, increased with longer incubation times (Fig 5H), suggesting that the cut in the aHC network acts as a dye-sink, allowing dye to be lost into the bath solution. To track true coupling coefficients, each equation was only fitted to the curve following the peak in cell fluorescence (closed markers Fig 6).

Methods 2 and 3 produced good fits of the cell fluorescence data at all timepoints (Fig 6B and 6C). Comparatively, Method 1 did not follow the curvature of the concentration series, resulting in a poorer fits at longer incubation times (Fig 6A, comparison of regression coefficients at one minute and ten minutes, $R^2$ = 0.93±0.01 vs. 0.85±0.02, $T(13)$ = 2.57, $p$ = 0.023). As expected, the space constant (μm) of dye-transfer in aHCs increased significantly with the duration of diffusion (Fig 6D, $F(4, 38)$ = 12.17, $p < 0.001$). This remained true when data were adjusted for cell-cell spacing (Fig 6G, Brown-Forscythe, $F(4, 20.6)$ = 40.80, $p < 0.001$).

Estimation of the coupling coefficient $k_j$ of aHCs via Method 2 produced consistent results across all incubation times, when calculated both in terms of absolute distance (Fig 6E, $F(4,38)$ = 0.65, $p$ = 0.63) and normalised cell-spaces (Fig 6H, $F(4,38)$ = 0.78, $p$ = 0.54). The mean coupling coefficient $k_j$ of aHCs across all conditions was 1.9 x $10^{-6}$ $cm^2.s^{-1}$ or 0.038 $cells^2.s^{-1}$. As with Method 2, Method 3 produced consistent estimation of the effective diffusion coefficient $De$ across all incubation times, when calculated both in terms of absolute distance (Fig 6F, $F(4,38)$ = 0.65, $p$ = 0.63) and normalised cell-spaces (Fig 6I, $F(4,38)$ = 0.45, $p$ = 0.77). The mean effective diffusion coefficient $De$ of aHCs across all conditions was 1.86 x $10^{-6}$ $cm^2.s^{-1}$ or 0.076 $cells^2.s^{-1}$. Across all methods, adjusting for the mean cell-cell spacing resulted in fewer outliers than if units were presented in terms of absolute distance (compare Fig 6D–6F and Fig 6G–6I).

## Experiment 2. Analysis of variable tracer movement at a fixed time point

Example images depicting the extent of dye diffusion of aHCs in retinae incubated in 50μM MFA and 250μM MFA are depicted in Fig 7A and 7B respectively. All three methods produced adequate dose-response curves, but with deviations in the calculated $IC_{50}$ between approaches (Fig 7C–7H). A large disparity existed between Method 1 and Methods 2 and 3 (all normalised to cell-cell separation) at 150μM MFA, with Method 1 displaying a significantly smaller mean reduction in cell-coupling compared to both Method 2 (41.1 ± 10.3% vs. 77.1 ± 16.8, $T(3)$ = 7.68, $p$ = 0.005) and Method 3 (41.1 ± 10.3% vs. 78.1 ± 12.6%, $T(3)$ = 8.41, $p$ = 0.004). This was consistent regardless of whether or not distance was normalised to cell-spacing (Table 3, Method 1 in μm). When the extent of coupling with 150μM MFA concentration was compared to the baseline MFA concentration of 50μM, Method 1 displayed a slightly lower statistical power than Methods 2 and 3 (Table 3). At high concentrations of MFA (250μM), the reduction in coupling compared to the baseline condition of 50 μM MFA when estimated using Method 1 (normalised to cell-cell separation) produced a significantly lower maximum reduction in cell-coupling than Methods 2 (60.6 ± 3.9% vs. 93.3 ± 1.1%, $T(3)$ = 8.56, $p$ = 0.003) and Method 3 (60.6 ± 3.9% vs. 94.3 ± 1.4, $T(3)$ = 8.40, $p$ = 0.004). This difference was also evident when data was presented as absolute distance (Table 3, Method 1 in μm).

## Experiment 3. Identification of coupling rates in different retinal cells subtypes

Amacrine cells expressing the enzyme neuronal nitric oxide synthase (nNOS) were also identified in tissues from Experiment 2 (Fig 8A). Method 2 and Method 3 were used to measure the coupling coefficient $k_j$ ($cells^2.s^{-1}$) and non-normalised diffusion coefficient $De$ ($cm^2.s^{-1}$) for both type-1 and displaced nNOS amacrine cells (Fig 8A). The rate of dye-transfer across each cell-cell connection ($k_j$) was significantly faster between displaced nNOS amacrine cells compared to type-1 nNOS amacrine cells (Fig 8G, 7.91 ± 1.37 x $10^{-4}$ vs. 2.69 ± 0.44 x $10^{-4}$ $cells^2.s^{-1}$, $T(21)$ = 3.773, $p$ = 0.001). However, the mean distance between type-1 nNOS cells was significantly larger than displaced nNOS cells (Fig 8F, 177.4 ± 7.8 vs. 74.5 ± 4.5 μm, $T(21)$ = 10.72, $p < 0.001$). As such, the overall rate of dye diffusion through the retina ($De$) was significantly

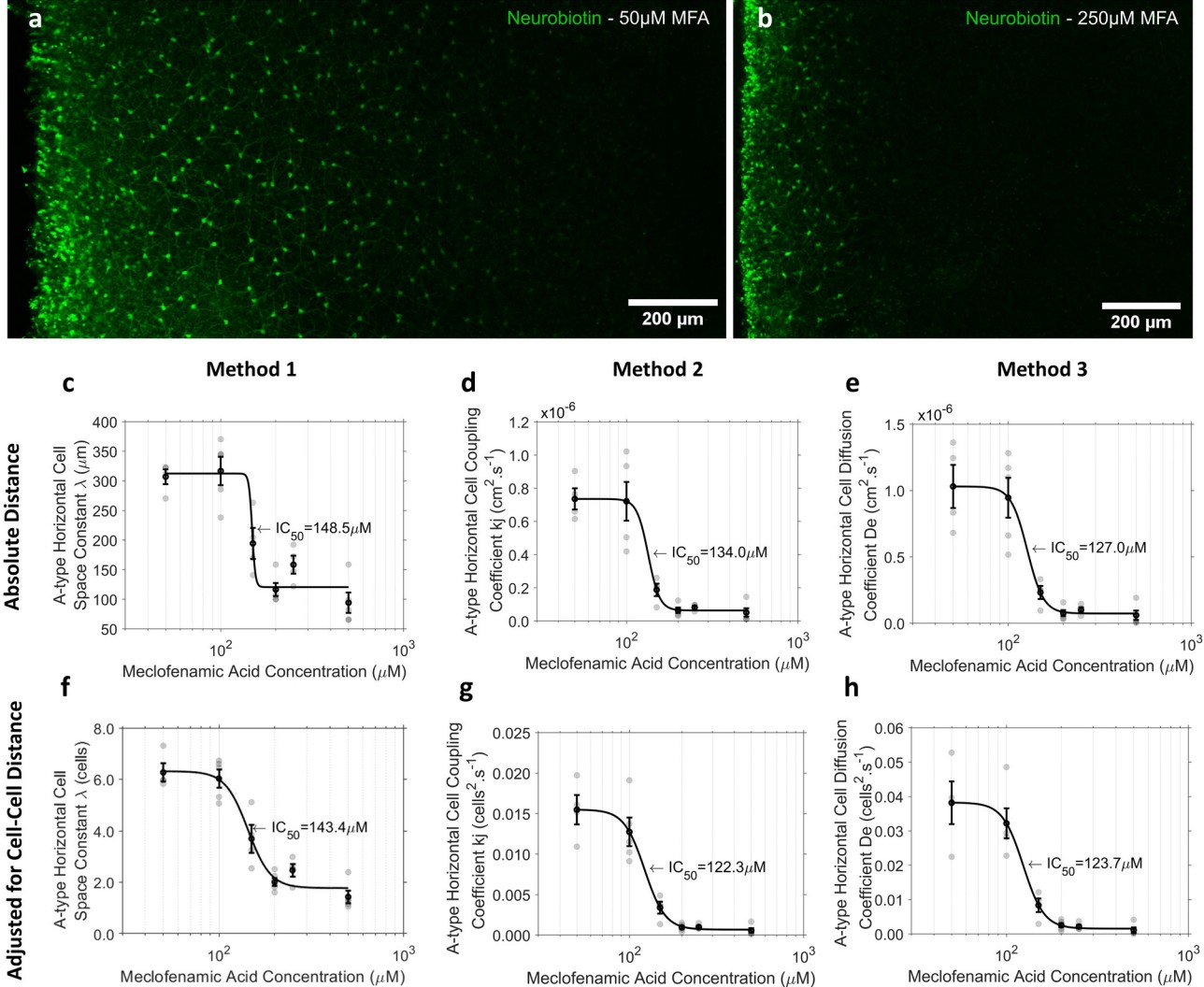

**Fig 7. Comparison of three analytical methods for describing dose-response data for a-type horizontal cell (aHC) coupling and meclofenamic acid (MFA) concentration *in vitro*.** The relative diffusion of Neurobiotin[TM] through HCs in tissue incubated in **(a)** 50μM MFA and **(b)** 250μM MFA. Column 1 **(c, f)** contains data analysed using Method 1, column 2 **(d, g)** shows the same data analysed using Method 2 and column 3 **(c, f, i)** shows the same data analysed using Method 3. The first row of plots **(c, d, e)** contains the mean ± SE (black marker, black error bars) and individual measurements (grey markers) for each measure of cell-coupling obtained via Methods: **(c)** 1, **(d)** 2 and **(e)** 3. Row 2 **(f, g, h)** contains the mean ± SE (black marker, black error bars) and individual measurements (grey markers) for each measure of cell-coupling normalised to the mean cell-cell spacing between adjacent aHCs, obtained via Methods **(f)**1, **(g)** 2 and **(h)** 3.

**Table 3. Reduction in cell coupling induced by the coupling inhibitor MFA, presented as a percentage reduction from a low-dose baseline concentration.** *p*-values indicate the statistical difference in calculated normalised effect sizes compared to Method 1.

| | | 150μM Vs. 50 μM MFA | | | 250μM Vs. 50 μM MFA | | |
|---|---|---|---|---|---|---|---|
| | Units | Normalised Effective Reduction (%) | Statistical Power | *p*-value | Normalised Effective Reduction (%) | Statistical Power | *p*-value |
| Method 1 ($\lambda$) | μm | 36.7 ± 9.5 | 0.893 | | 48.4 ± 5.0 | 1.000 | |
| | cells | 41.1 ± 10.3 | 0.911 | | 60.6 ± 3.9 | 1.000 | |
| Method 2 ($k_j$) | cm$^2$.s$^{-1}$ | 74.5 ± 10.1 | >0.999 | 0.004 | 88.9 ± 1.3 | 1.000 | 0.011 |
| | cells$^2$.s$^{-1}$ | 77.1 ± 16.8 | >0.999 | 0.005 | 93.3 ± 1.1 | 1.000 | 0.003 |
| Method 3 (*De*) | cm$^2$.s$^{-1}$ | 77.5 ± 12.6 | 0.983 | 0.003 | 90.1 ± 1.8 | >0.999 | 0.012 |
| | cells$^2$.s$^{-1}$ | 78.1 ± 12.6 | 0.966 | 0.004 | 94.3 ± 1.4 | >0.999 | 0.004 |

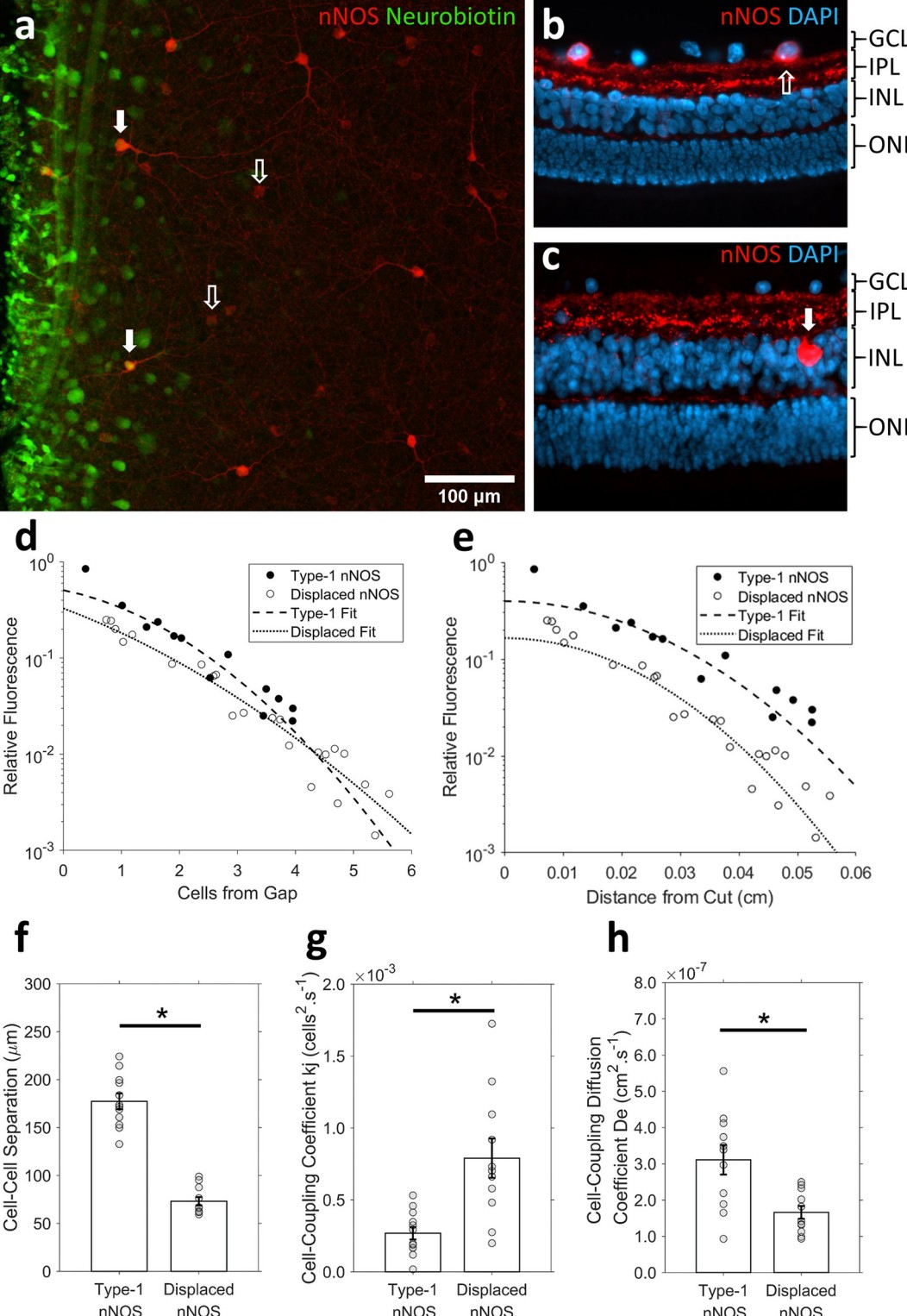

**Fig 8. Identification of amacrine cells expressing neuronal nitric oxide synthase loaded with molecular tracer via cut-loading. (a)** Image created from z-stack slices spanning from the inner nuclear layer to the ganglion cell layer, flattened using the z-project function in Fiji. Open arrows, type-1 nNOS amacrine cells; solid arrows, displaced nNOS amacrine cells. **(b, c)** Vertical retinal sections labelled for nNOS (red) and DAPI (blue). **(b)** Open arrows highlight type-1 nNOS amacrine cells located in the inner nuclear layer (INL), **(c)** solid white arrows highlight displaced nNOS amacrine cells located in the

ganglion cell layer (GCL). **(d, e)** Fluorescent intensity of Neurobiotin™ in cut-loaded type -1 and displaced nNOS amacrine cells analysed using **(d)** Method 2 (fitted with Eq 6); or **(e)** Method 3 (fitted with Eq 9). **(f-g)** Comparison of nNOS type-1 and nNOS displaced amacrine cells in their mean: **(f)** cell density, **(g)** cell-coupling coefficients *kj*, and **(h)** relative diffusion coefficients. Error bars are SEM. * $p < 0.05$.

faster in the type-1 nNOS cell network than through the displaced nNOS amacrine cell network (Fig 8H, $3.11 \pm 0.41$ x $10^{-4}$ vs. $1.73 \pm 0.18$ x $10^{-4}$ cm$^2$.s$^{-1}$, $T(21) = 3.00$, $p = 0.007$).

## Discussion

In the present study, we demonstrate the versatility of the cut-loading method and highlight several important considerations that should be taken into account when analysing cut-loading data.

Although the standard analysis protocol has several limitations, many of these can be addressed with some simple modifications to the analysis protocol (Method 1). The variation arising from changes in cell-density between tissues/sampling area (limitation 1) can be addressed by dividing the absolute distance by the mean cell-cell distance, resulting in the units of the space constant being cells as opposed to μm. Limitation 2 related to non-specific fluorescence can be improved by measuring the mean soma fluorescence of each cell, rather than the mean fluorescence of an area. This modification also addresses the disproportionally high fluorescence measured at the cut by the standard technique, resulting from cells other than HCs filling with dye. By measuring the soma fluorescence, the data may still be normalised based on the fluorescence of cells located at the cut, but this data can then be excluded from the fit. As others have previously demonstrated, these cells may be identified by including high molecular weight Rhodamine-dextran in the cut-loading solution as these would load the cells located at the cut but would not readily pass into the tissue [34]. Although these simple modifications improve the validity of using the space constant as a means of describing cell-coupling in cut-loading, they do not allow for direct comparison with studies involving dye-injection.

Methods 2 and 3 described the decay in cell fluorescence more accurately than Method 1. Additionally, as methods 2 and 3 utilised incubation time to determine the rate of dye transfer, as opposed to the distance of dye travel (method 1), they produced consistent estimations of $k_j$ and *De* respectively when applied across a range of incubation times. Methods 2 and 3 were also more sensitive to smaller changes in cell-coupling at a fixed time point than Method 1, producing approximately twice the effect size without affecting high statistical power in the 150μM MFA group and approximately 1.5 times the maximum effect size at 250μM MFA (Table 3).

Depending on the associated pharmacological mechanism, gap-junction inhibitors may produce either partial or complete blockage of gap-junctions [46, 47]. This efficacy, along with solubility, toxicity and effective concentration is important for determining the appropriate drug for experimental applications. Therefore, although Method 1 produced a similar IC$_{50}$ value to Methods 2 and 3, its underrepresentation of the maximum dose-response of MFA make Methods 2 and 3 are more appropriate for this application.

### Comparison of coupling ($k_j$) and diffusion (*De*) coefficients between techniques

Our estimation of the effective diffusion coefficient *De* of Neurobiotin™ through coupled aHCs was similar to that previously measured in Rabbit retinae ($1.86$ x $10^{-6}$ cm$^2$.s$^{-1}$ vs. $2.56$ x $10^{-6}$ cm$^2$.s$^{-1}$) [38]. However, our measured value for the coupling coefficient $k_j$ was substantially higher ($0.038$ cells$^2$.s$^{-1}$ vs. $0.003$ cells$^2$.s$^{-1}$). This difference may reflect differences in the

extent of light/dark adaptation between experimental conditions [3, 48], or interspecies variation. However, considering our estimation of the diffusion coefficient was similar to that in the Rabbit, it is likely that this difference arises from our mathematical model. One explanation is that the mathematical model used in Eq (6), presumes no rise to maximum fluorescence in the first cell of the coupled network, as this duration could not be known from fixed time-point analysis such as that performed in cut-loading. This is opposed to the method used by Mills and Massey, where a continuous injection of a known rate and duration for $c_1$ is subsequently incorporated into the analysis [38]. However, when one of their example Rabbit traces loaded via microinjection (30 minute incubation) was reanalysed using Eq (6) (presuming no rise to maximum, with no adjustment for cell geometry), $k_j$ values remained four times higher in Guinea pig HCs (0.038 cells$^2$.s$^{-1}$ vs. 0.0095 cells$^2$.s$^{-1}$). However, by adjusting for the geometric path of dye-travel through a triangular lattice from a single cell-source, Eq (12) [49], the coupling coefficient $k_j$ from cut-loaded Guinea pig HCs was similar to estimates from Rabbit HCs investigated using single cell micro-injection (0.038 cells$^2$.s$^{-1}$ vs. 0.019 cells$^2$.s$^{-1}$).

$$\frac{dC_1}{dx} = 6k_j(C_2 - C_1) - k_s(C_1 V_1) \tag{12}$$

$$\frac{dC_n}{dx} = k_j\big(2.5C_{n+1} + 1.5C_{n-1} - 4C_n\big) - k_s(C_n V_n)$$

**Uses of each methodological approach.** The application of the methods detailed in the present study would depend on the information desired. Method 1 is a useful technique for comparing the absolute spread of molecular tracer within different cell networks over a fixed time. When the cell-density and tiling geometry are accounted for, as in Eq (6) of Method 2, the rate of dye-transfer across each cell-cell connection ($k_j$) may be estimated. This method therefore allows for the simultaneous comparison of the relative permeability of gap-junctions present between different cell-types. Comparatively, Method 3 calculates the rate of passive diffusion of tracer through a cell network, away from the initial cut. This is of use when the absolute spread of a signal/tracer across the retina through a specific cell network is of interest. This may be normalised to the cell-cell separation if the relative decay of a signal across adjoining cells is of interest, otherwise, the absolute distance of the signal/tracer spread across the retina may be obtained from non-normalised data. For example, the rate of dye-transfer between individual displaced nNOS amacrine cells ($k_j$) is faster than type-1 nNOS amacrine cells. However, as displaced nNOS cells are more densely distributed in the retina, the absolute distance that the signal will reach in a given time (*De*) is less than that in the less dense, type-1 nNOS cell network. Each of these analysis techniques may also be applied to scrape loading techniques (on which cut-loading is originally based), as dye-diffusion occurs from a line source, feeding coupled cells arranged in a triangular lattice [50].

**Measuring specific cell-types in highly heterogeneous cell layers.** A limitation of the cut-loading method, is that it cannot be used to delineate the path of dye-travel, as information detailing whether cells are homologously or heterologously coupled is not attainable. However, the advantage of the cut-loading technique is that it may be expanded to measure cells interspersed in the inner nuclear and retinal ganglion cell layers by counter labelling cut-loaded tissues with antibodies of specific cell markers and measuring the fluorescence of individual cell soma. This technique is limited by the number of compatible fluorophores for secondary antibodies but offers considerably advantage over dye injected into an individual cell as coupling coefficients can be simultaneously assessed for a range of different cell types in the same tissue after a particular experimental manipulation.

## Conclusion

Cell-coupling is important to many visual circuits within the retina. However, the functional significance of cell-coupling between many retinal cell-types as well as how these circuits are modulated or otherwise altered in disease, remains to be established. In spite of its simplicity, cut-loading enables cell-coupling data to be obtained simultaneously through different coupled networks across all layers of the retina. By employing the techniques described here, this analysis can be expanded to include assessments of the relative coupling coefficient ($k_j$) and effective diffusion coefficient ($De$) of molecular tracer through the retina.

## Supporting information

**S1 File. Cut-loading analysis MATLAB file.** A simple MATLAB code for calculating the normalised and non-normalised coupling and diffusion coefficients (based on Eqs 6 and 10 respectively) for a cut-loaded network of coupled cells arranged in a triangular lattice. (M)

## Author Contributions

**Conceptualization:** Sally A. McFadden.

**Data curation:** William E. Myles.

**Formal analysis:** William E. Myles.

**Funding acquisition:** Sally A. McFadden.

**Investigation:** William E. Myles.

**Methodology:** William E. Myles.

**Project administration:** Sally A. McFadden.

**Resources:** Sally A. McFadden.

**Software:** William E. Myles.

**Supervision:** Sally A. McFadden.

**Validation:** William E. Myles.

**Writing – original draft:** William E. Myles.

**Writing – review & editing:** William E. Myles, Sally A. McFadden.

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
