## [Decision Letter · Decision Letter 0]

7 Apr 2022

PONE-D-22-03044Analytical Methods for Assessing Retinal Cell Coupling Using Cut-LoadingPLOS ONE

Dear Dr. Myles,

Thank you for submitting your manuscript to PLOS ONE. After careful consideration, we feel that it has merit but does not fully meet PLOS ONE’s publication criteria as it currently stands. Therefore, we invite you to submit a revised version of the manuscript that addresses the points raised during the review process.

Specifically, we strongly recommend to follow reviewers' comments, including the suggestion to include a gap junction-impermeable large molecular weight dye (such as rhodamine dextran) or other gap junction permeable dyes (such as lucifer yellow). These latter studies will permit assessment of coupling coefficients using a positive and negatively charged molecules. Moreover, reference #11 was used to support the description of Cx expression in the retina. Original references should be cited instead, as for instance:

Connexin36 is required for gap junctional coupling of most ganglion cell subtypes in the mouse retina. Pan F, Paul DL, Bloomfield SA, Völgyi B.J Comp Neurol. 2010 

Coupling between A-type horizontal cells is mediated by connexin 50 gap junctions in the rabbit retina. O'Brien JJ, Li W, Pan F, Keung J, O'Brien J, Massey SC.J Neurosci. 2006 

Expression of connexins 36, 43, and 45 during postnatal development of the mouse retina. Kihara AH, Mantovani de Castro L, Belmonte MA, Yan CY, Moriscot AS, Hamassaki DE.J Neurobiol. 2006 

Regarding alterations in GJ coupling by light/dark, studies on regulation of Cx should be cited, as for instance:

Prolonged dark adaptation changes connexin expression in the mouse retina. Kihara AH, de Castro LM, Moriscot AS, Hamassaki DE.J Neurosci Res. 2006 

 Please submit your revised manuscript by May 14 2022 11:59PM. If you will need more time than this to complete your revisions, please reply to this message or contact the journal office at plosone@plos.org. Please include the following items when submitting your revised manuscript:

We look forward to receiving your revised manuscript.

Kind regards,

Alexandre Hiroaki Kihara, Ph.D.

Academic Editor

PLOS ONE

Journal Requirements:

3. Please ensure that you refer to Figure xxxxx in your text as, if accepted, production will need this reference to link the reader to the figure.

Reviewers' comments:

Reviewer's Responses to Questions

**Comments to the Author**

1. Is the manuscript technically sound, and do the data support the conclusions?

Reviewer #1: Yes

Reviewer #2: Yes

Reviewer #3: Partly

2. Has the statistical analysis been performed appropriately and rigorously? 

Reviewer #1: Yes

Reviewer #2: Yes

Reviewer #3: Yes

3. Have the authors made all data underlying the findings in their manuscript fully available?

Reviewer #1: Yes

Reviewer #2: Yes

Reviewer #3: Yes

4. Is the manuscript presented in an intelligible fashion and written in standard English?

Reviewer #1: Yes

Reviewer #2: Yes

Reviewer #3: Yes

5. Review Comments to the Author

Reviewer #1: The manuscript by Myles and McFadden compares different methods to quantify the diffusion of a biotinylated tracer through gap junctions in retina horizontal cells following cut-loading. The three techniques used are well described in the manuscript. These techniques have been published previously by different groups, but never compared. The reported effects of MFA are consistent with the literature. The figures are of great quality and informative. This is a nice piece of work, clearly written, and overall a nice comparison of the different techniques established by others. I have only minor comments.

Minor:

- Line 12: Define A-type HCs the first time you use it (lines 12 and 119)

- Line 32: not all retinal cells are coupled (e.g. rod bipolar cells); consider replacing the start of the sentence with “Most cells of the retina are coupled….” Actually I don’t like that sentence because correlated spiking is observed only between RGCs and not between photoreceptors or horizontal cells… Consider merging sentence 1 and 2.

- Line 42: based

- Line 48: please add Bloomfield and Volgyi, 2009, Nature Neurosciences Reviews, here. I am not sure of Cx43 in RGCs. I don’t want to be picky but HCs express Cx50 (A-type) OR Cx57 (B-type), usually only 1 Cx is expressed per cell type, hence that sentence should contain OR instead of AND.

- Line 50: ref 12 is for Cx36, there should be a general review cited here for the plasticity of connexins. Usually, they are regulated by phosphorylation, but not always. There should be a review from John O’brien here as well.

- Lines 49-55: Mills and Massey (1995) Nature should be cited in this paragraph

- Lines 59-60: refs 16 and 17 are from other brain areas. I suggest you cite references in the retina. For cell pairs: DeVries et al. (2001) Current Biology; Li et al. and Schnapf (2012) J Neuroscience; Jin et al. and Ribelayga (2020) Science adv. Or another ref from the DeVries, Schnapf, or Ribelayga labs. For tracer injections: a reference from the Bloomfield lab, the Ribelayga lab or the Mangel lab. For cut-loading, the ref is correct, might want to add one or two from the O’brien lab as well (Li et al., 2013 and 2009, J. Neuroscience).

- Lines 79-80. Yes they have! See Li, Chuang, and O’Brien (2009) J Neurosicence; Li et al. and O’Brien (2013) J Neuroscience.

- Lines 90-92: see comment above, check O’Brien’s publications.

- Line 92: provide a ref for Fick’s second law

- Line 117: replace “for a single duration” by “for 20 minutes”

- Line 158: diffuse instead of disperse

- Lines 171+173: from exps 1 and 2 or only from exp 2?

- Table 2: provide the source of anti-Calbindin.

- Line 187: if you collapse the entire z-stack (spans throughout the retina), how can you be sure that you are measuring fluorescence in a single cell? The fluorescence from the cells underneath would add up… This needs clarification here. Also see next comment.

- Line 212: what do you call “background fluorescence”

- Line 310: aHC could be used from the first time a-type horizontal cells are mentioned in the text (and after line 310)

- Lines 312-319: please add details, difficult to follow which red curve you are referring to (there are 2 on the figure)

- Lines 312-319: are you measuring tracer coupling in a-type and b-type networks at the same time? The diffusion of Neurobiotin has probably different kinetics in each network (see Mills and Massey, 1998). Figure 5 shows that without calretinin labelling, it is very difficult to distinguish the 2 types based on the morphology of the soma.

- Lines 491-491: which HC type?

Reviewer #2: This manuscript describes analysis of cut-loading data in attempt to bring a greater degree of analytical analysis to the resulting data. I believe the authors have achieved this goal. I have a few minor issues that need addressing.

1. The first example of cut-loaded retina provided (Fig. 4a) seems very low contrast. I had to turn up the brightness on the image to see clearly the cells labelled distant from the cut. It is also concluded that the labelled cells are a-type HCs. How do they know this? Later in the manuscript (Fig. 5) there is discussion of using calbindin (to distinguish HCs from other retinal neurons) and morphology (to distinguish a- from b-type HCs). Perhaps this needs to be introduced earlier so that it is clear how the authors know the cells in Fig. 4a are a-type HCs.

2. Fig. 5a-e: Stated as being a-type HCs. How is this known?

3. Regarding Fig. 5f and g: I guess the appearance of the soma and primary neurites of the representative a- and b-type HCs (Fig. 5f) corresponds to previous descriptions of guinea pig HCs (Peichl & Gonzalez-Soriano, 1994). But it would be even more convincing if the axon of the b-type cells could be seen. Fig. 5g shows cells double-labelled for Neurobiotin and calbindin. But there seem to be many cells that are only Neurobiotin (NB) labelled. What are these cells? So many NB+ cells visible at the same depth as HC (that are not calbindin+?) would confound the analysis. I think I am missing something important here that needs to be clarified in the manuscript.

4. The analysis illustrated in Fig. 6 highlights cells immediately adjacent to the cut that had reduced fluorescence. (It is suggested that if very near the cut, some cells may lose NB.) These are indicated in the graphs as unfilled circles -- but this detail is not provided in the Figure legend. I guess I can see these dimmer cells in Fig. 6a-e, but not in Fig. 4a. Indeed, in my brightness-enhanced version of Fig 4a there seem to be many cells adjacent to the cut, many more than can bee seen distance from the cut. Why?

5. Although statistically different, are the differences in the IC50 for MFA really biologically significant?

6. I can see that the improved analysis could be quite useful when it comes to comparing coupling between different cell types (e.g. Fig. 8), but given the often dramatic changes in gap junction coupling seen with changing ambient illumination or treatments with neuromodulators (e.g. dopamine, nitric oxide, retinoic acid, etc.) I am less certain that the analysis would add much. Perhaps the authors would like to include such a consideration in the Discussion.

7. Very minor point, but in the very first sentence of the Introduction it is emphasized how gap junction coupling can result in correlated spiking. But since the manuscript (and the first sentence!) is about retina, where many (most?) of the cells do not spike, perhaps this is not the best way to start!

Reviewer #3: The paper by Myles and McFadden used the cut-loading method and demonstrated its utility to understand gap junction-coupled networks in the retina. The authors analysed coupled networks using three different methods and highlight the importance of these methods. The data are sound and the paper is well-written.

My main concern is that while cut-loading with neurobiotin and subsequent analysis is useful for cell types that are extensively coupled (such as horizontal cells), its utility to understand gap junctional coupling in cell types such as ganglion cells, bipolar cells and amacrine cells is extremely limited. These retinal neurons have multiple subtypes (e.g. there are greater than 10 types of amacrine cells depending on the species), and not all of these subtypes express gap junctions. So it is unclear what this method would offer over conventional dye-injection methods, where tracers are introduced into a single cell.

Second, different neuronal subtypes in the retina are coupled to each other (e.g. amacrine cells and certain RGCs). Also, coupling exists between glial cells in the retina, and may even be unidirectional, as shown by Robinson et al (1993) and Newman and others (1997). How will the cut-loading method establish which of these cell types was first loaded? (On a side note, Cx43 is expressed in glial cells and not in neurons as stated in the Introduction)

The paper would have improved considerably, if the authors chose to include a gap junction-impermeable large molecular weight dye (such as rhodamine dextran) or other gap junction permeable dyes (such as lucifer yellow). These latter studies will permit assessment of coupling coefficients using a positive and negatively charged molecules.

6. PLOS authors have the option to publish the peer review history of their article (what does this mean?). If published, this will include your full peer review and any attached files.

Reviewer #1: No

Reviewer #2: No

Reviewer #3: No

---

## [Author Response · Author response to Decision Letter 0]

16 May 2022

- Line 12: Define A-type HCs the first time you use it (lines 12 and 119)

 This has now been changed 

 Line 14: ‘to assess coupling strength in axonless ‘a-type’ horizontal cells (aHCs).’

Line 123: ‘in Experiment 1. In both experiments, axonless ‘a-type’ horizontal cells (aHC) were selected as a model system for analyses.’

- Line 32: not all retinal cells are coupled (e.g. rod bipolar cells); consider replacing the start of the sentence with “Most cells of the retina are coupled….” Actually I don’t like that sentence because correlated spiking is observed only between RGCs and not between photoreceptors or horizontal cells… Consider merging sentence 1 and 2.

This has now been changed 

Line 35: ‘Most of the cells of the retina are extensively linked by intercellular gap-junctions that allow the intercellular passage of ions and small molecules (typically up to 1000 Da) between pairs or networks of coupled neurons.’

- Line 48: please add Bloomfield and Volgyi, 2009, Nature Neurosciences Reviews, here. I am not sure of Cx43 in RGCs. I don’t want to be picky but HCs express Cx50 (A-type) OR Cx57 (B-type), usually only 1 Cx is expressed per cell type, hence that sentence should contain OR instead of AND. 

The Bloomfield and Volgyi review has been cited (citation 6). The use of ‘and’ has been replaced by ‘or’ when identifying connexin subtypes observed in retinal cells. Additionally, in line with the editor’s comments, references have been added for each individual connexin isoform that has been observed in the retina. 

Line 48: ‘The cells of the retina express several connexin isoforms (6). Retinal pigment epithelial cells express Cx43, cone and rod photoreceptors express Cx36 (11, 12), horizontal cells express Cx50 (13) or Cx57 (14), bipolar and amacrine cells express Cx36 or Cx45, and retinal ganglion cells express Cx30.2 (15), Cx36 (16), or Cx45 (17).’

- Lines 49-55: Mills and Massey (1995) Nature should be cited in this paragraph

 This citation has been added to this section (citation 26)

Line 53: ‘In the retina, this occurs in response to the release of light-mediated neuromodulators such as dopamine [23, 24] and nitric oxide [25, 26], facilitating in the switching between light and dark processing pathways.’

- Lines 59-60: refs 16 and 17 are from other brain areas. I suggest you cite references in the retina. For cell pairs: DeVries et al. (2001) Current Biology; Li et al. and Schnapf (2012) J Neuroscience; Jin et al. and Ribelayga (2020) Science adv. Or another ref from the DeVries, Schnapf, or Ribelayga labs. For tracer injections: a reference from the Bloomfield lab, the Ribelayga lab or the Mangel lab. For cutloading, the ref is correct, might want to add one or two from the O’brien lab as well (Li et al., 2013 and 2009, J. Neuroscience).

These references have now been included in the respective sections

Line 61: ' Methods for assessing cell-cell coupling include simultaneous electrical recordings from cell pairs [27-29], intracellular microinjection of molecular tracers [30, 31], and cut-loading [32, 33].’ 

- Line 50: ref 12 is for Cx36, there should be a general review cited here for the plasticity of connexins. Usually, they are regulated by phosphorylation, but not always. There should be a review from John O’brien here as well.

The wording of this sentence has been changed accordingly

Line 52: ‘The conductance of gap-junctions may be transiently regulated by the phosphorylation of serine residues in the component connexin proteins [20-22].’

- Lines 79-80. Yes they have! See Li, Chuang, and O’Brien (2009) J Neurosicence; Li et al. and O’Brien (2013) J Neuroscience.

Thank you for pointing out this omission. This sentence has been changed and references to the respective studies have been included

Line 83: ‘in which only single time-point data is available [38], however, few studies have integrated this analytical approach as part of the cut-loading protocol [33, 39].’ 

- Line 117: replace “for a single duration” by “for 20 minutes” 

This has been corrected

Line 120: ‘In Experiment 2, retinae were cut-loaded with molecular tracer and incubated for 20 minutes.’ 

- Line 158: diffuse instead of disperse

This has been corrected

Line 168: ‘The tissue was returned to the Ames solution and the NeurobiotinTM dye allowed to diffuse through the cell network.’ 

- Table 2: provide the source of anti-Calbindin.

 This omission has been corrected 

‘ 

Calbindin D-28K AB1778 Rabbit Sigma-Aldrich 1:400 Overnight (4oC)

‘

- Line 187: if you collapse the entire z-stack (spans throughout the retina), how can you be sure that you are measuring fluorescence in a single cell? The fluorescence from the cells underneath would add up… This needs clarification here. Also see next comment.

We collected z-stacks spanning through the entire retina, however we only used 10 slices that contained either the aHCs or nNOS amacrine cells. We have clarified this in the text.

Line 195: ‘Composite images used for analysis were created from 10 slices selected from the z-stack using the sum-slices z-project function in Fiji. For aHCs these 10 slices spanned from the boundary of the outer plexiform layer and the inner nuclear layer.’ 

- Line 212: what do you call “background fluorescence”

 We have included a more detailed description in the text

Line 223: ‘The background fluorescence of the retinal tissue was measured for each composite image in a region approximately 1500µm from the cut which contained no fluorescing cells and was subtracted from absolute fluorescence measurements.’

Line 310: aHC could be used from the first time a-type horizontal cells are mentioned in the text (and after line 310)

 This has been corrected

Line 323: ‘An example cut-loaded retinal image of NeurobiotinTM filled aHC spanning 1.5 mm from the cut is presented in Fig. 4a’

- Lines 312-319: please add details, difficult to follow which red curve you are referring to (there are 2 on the figure)

Thank you for pointing this out, the pink curve has been replaced with a blue curve to better contrast with the existing red curve. 

Line 334: ‘An alternative method to the standard protocol (Method 1) was based on the mean fluorescence for each aHC soma (Fig 4f, blue dashed line fit to grey closed markers)’

- Lines 312-319: are you measuring tracer coupling in a-type and b-type networks at the same time? The diffusion of Neurobiotin has probably different kinetics in each network (see Mills and Massey, 1998). Figure 5 shows that without calretinin labelling, it is very difficult to distinguish the 2 types based on the morphology of the soma.

Thank you for your comment. As was observed by Mills and Massey, we see very little diffusion of neurobiotin through coupled B-type horizontal cells in our cut-load preparations versus A-type horizontal cells. To this effect, we are typically only able to observed B-type horizontal cells within 100-200um from the cut and this usually requires higher magnification than was used in images a-e. We have added the fluorescence data of b-type HCs to Fig 4f, to highlight this distinction. We have also added two additional images to Fig 4. Demonstrating Calbindin labelling of a-type and b-type HCs, which are both filled with dye immediately adjacent to the cut (Fig 4d), and anther showing Calbindin labelling of a-type and b-type HCs with only a-type HCs filled with dye when viewed 500um away from the cut (Fig 4e). We have provided clarification of this in the text. 

Line 329: ‘This layer contained both highly coupled aHCs as well as the less coupled axon-bearing ‘b-type’ HCs (bHCs). These subtypes were easily distinguishable based on their distinct cellular morphology (Fig 4c). Both subtypes co-labelled with calbindin (Fig 4d), however, NeurobiotinTM was only observable in bHCs within 200µM from the cut, whereas NeurobiotinTM dye was observed in aHCs up to 1500µM from the cut (Fig 4a, e).’

Reviewer #2 

The first example of cut-loaded retina provided (Fig. 4a) seems very low contrast. I had to turn up the brightness on the image to see clearly the cells labelled distant from the cut. It is also concluded that the labelled cells are a-type HCs. How do they know this? Later in the manuscript (Fig. 5) there is discussion of using calbindin (to distinguish HCs from other retinal neurons) and morphology (to distinguish a- from b-type HCs). Perhaps this needs to be introduced earlier so that it is clear how the authors know the cells in Fig. 4a are a-type HCs.

Thank you for highlighting these issues. The image has been cropped and enlarged to allow for easier viewing. Additionally, we have moved the distinction of A-type and B-type cells forward in the results section and have included the morphological comparison image to Fig 4 (now Fig 4c). Additionally, we have included two new images of Calbindin labelling to Fig 4 (Fig 4d, e). Calbindin labels both subtypes of HC. Fig 4d shows labelling of both bHC and aHCs at the cut, all of which have been loaded with neurobiotin dye. However, because bHCs are far less coupled than aHCs (Mills and Massey, 1998), they can only be seen within the first 100 – 200 μm from the cut and they are rarely observable at low (10x) magnification as we used in Fig 4a. In Fig 4e (that was taken ~500 μm from the cut) both aHCs and bHCs label with Calbindin, but only aHCs contain neurobiotin dye (green) due to the far greater coupling observed in aHCs versus bHCs. This accounts for your observation of additional cells adjacent to the cut in Fig 4a (see our response to comment 4 for more details). We have also added the measurements of the fluorescence of bHCs to Fig 4f to better demonstrate the distinction between bHC and aHC coupling. 

Line 323: ‘An example cut-loaded retinal image of NeurobiotinTM filled aHCs spanning 1.5 mm from the cut is presented in Fig 4a, the associated fluorescence data in Fig 4b, and the same data translated onto a log scale in Fig 4c. The exponential decay function from the standard cut-loading analysis protocol, equation (1), produced a poorer fit to the data in regions closest to and furthest away from the cut (red dashed line in Fig 4c) but overall was well fitted (mean R2 of fit ± S.D = 0.89 ± 0.05, n = 44 cuts). This layer contained both highly coupled aHCs as well as the less coupled axon-bearing ‘b-type’ HCs (bHCs). These subtypes were easily distinguishable based on their distinct cellular morphology (Fig 4c). Both subtypes co-labelled with calbindin (Fig 4d), however, NeurobiotinTM was only observable in bHCs within 200µM from the cut, whereas NeurobiotinTM dye was observed in aHCs up to 1500µM from the cut (Fig 4a, e).

An alternative method to the standard protocol (Method 1) was based on the mean fluorescence for each aHC soma (Fig 4f, blue dashed line fit to grey closed markers) and had the advantage of excluding background fluorescence that was not relevant to the cells of interest and excluding fluorescence from dye-loaded bHCs, observed immediately adjacent to the cut (Fig 4f, purple dashed line fit to purple markers).’

2. Fig. 5a-e: Stated as being a-type HCs. How is this known?

The cells in these images were identified as aHCs based on morphology (aligning with Fig 4c) and coupling characteristics. Due to the limited coupling of bHCs, these cells are rarely observable after short incubation times (1 – 5 mins) and even after 10 – 20 minutes, these cells have a low fluorescence and can typically only be seen at higher magnification than that used for the images in this figure (the image in Fig 4a is an exception and was taken at the same 10x magnification as the images in Fig 5). These images were selected as they do not contain any bHCs 

3. Regarding Fig. 5f and g: I guess the appearance of the soma and primary neurites of the representative a- and b-type HCs (Fig. 5f) corresponds to previous descriptions of guinea pig HCs (Peichl & Gonzalez-Soriano, 1994). But it would be even more convincing if the axon of the b-type cells could be seen. Fig. 5g shows cells double-labelled for Neurobiotin and calbindin. But there seem to be many cells that are only Neurobiotin (NB) labelled. What are these cells? So many NB+ cells visible at the same depth as HC (that are not calbindin+?) would confound the analysis. I think I am missing something important here that needs to be clarified in the manuscript

Thank you for pointing out this issue. The Calbindin labelling was relatively poor in the image included the figure, with some HCs showing almost no signal. We have repeated our Calbindin labelling which demonstrates the labelling far more clearly. Calbindin labels both aHCs and bHCs, as shown in Fig 4d and e. As previously outlined, bHCs are poorly coupled and only fill with NB dye close to the cut location (Fig. 4d) whereas aHCs fill up to 2mm away from the cut following 20 minutes incubation and are the only HCs filled with dye from ~500 µm from the cut (Fig 4e). 

 4. The analysis illustrated in Fig. 6 highlights cells immediately adjacent to the cut that had reduced fluorescence. (It is suggested that if very near the cut, some cells may lose NB.) These are indicated in the graphs as unfilled circles -- but this detail is not provided in the Figure legend. I guess I can see these dimmer cells in Fig. 6a-e, but not in Fig. 4a. Indeed, in my brightness-enhanced version of Fig 4a there seem to be many cells adjacent to the cut, many more than can bee seen distance from the cut. Why?

Thank you for outlining this omission, we have added reference to the open markers in the figure legend. 

Line 421: “Open markers indicate cells adjacent to the cut with reduced fluorescence.”

Thank you for bringing this to our attention. There are more cells located along the boundary of the cut in figure 4 because this region contains both B-type (bHCs) and A-type (aHCs) horizontal cells that have filled with dye. Similar to what has been observed in Rabbit retinae (Mills and Massey, 1998), we observe far less coupling in bHCs than aHCs. To that extent, BHCs are typically only visible within 100-200 μm from the cut compared to the 1000-2000 μm for aHCs. Additionally, bHCs and far dimmer and are rarely visible with the 10x objective used in figure 4a, however, in this case they can be seen when the contrast is turned up. For example, Fig. 4c was taken with a 60x objective approximately 80 μm from the cut. To clarify this, additional images have been added to figure 4 showing Calbindin labelling of B-type and A-type HCs all filled with dye immediately adjacent to the cut (Fig 4d), this is contrasted to Fig 4e where Calbindin labels both B-type and A-type HCs ~500um away from the cut, however, only A-type HCs are filled with dye. This is due to aHCs being highly coupled and bHCs being far less coupled. Additionally, we have added data point for the decay in fluorescence of B-type HCs measured from Fig 4d to Fig 4f and have fit it using method 1. 

5. Although statistically different, are the differences in the IC50 for MFA really biologically significant?

The difference in MFA IC50 between the three analytical methods is small, and is unlikely to be biologically significant. However, there is a greater difference in the extent of coupling reduction due to MFA when calculated by method 1 compared to methods 2 and 3. This increased sensitivity would allow for greater distinction between the actions of pharmacological drugs on coupling networks as well as greater ability to distinguish the rate of coupling between different retinal cell types. 

6. I can see that the improved analysis could be quite useful when it comes to comparing coupling between different cell types (e.g. Fig. 8), but given the often dramatic changes in gap junction coupling seen with changing ambient illumination or treatments with neuromodulators (e.g. dopamine, nitric oxide, retinoic acid, etc.) I am less certain that the analysis would add much. Perhaps the authors would like to include such a consideration in the Discussion. 

Thank you for this valid critique. It is true that if the main outcome is to detect the EC/IC50 of pharmacological drugs on coupling, then methods 2 and 3 do not provide a significant advantage over method 1. However, methods 2 and 3 would provide a more sensitive and accurate assessment of the extent that drugs, neuromodulators or light reduce coupling. For example, methods 2 and 3 calculate a 93-94% decrease in coupling at >200μM MFA, however, this is under-represented by analysis method 1 at only 60%. This would be also true of the effects of light adaption and neuromodulator concentration on coupling. 

7. Very minor point, but in the very first sentence of the Introduction it is emphasized how gap junction coupling can result in correlated spiking. But since the manuscript (and the first sentence!) is about retina, where many (most?) of the cells do not spike, perhaps this is not the best way to start!

Thank you for pointing this out, this has now been corrected in the text 

Line 35: “Most of the cells of the retina are extensively linked by intercellular gap-junctions that allow the intercellular passage of ions and small molecules (typically up to 1000 Da) between pairs or networks of coupled neurons.”

Reviewer #3

My main concern is that while cut-loading with neurobiotin and subsequent analysis is useful for cell types that are extensively coupled (such as horizontal cells), its utility to understand gap junctional coupling in cell types such as ganglion cells, bipolar cells and amacrine cells is extremely limited. These retinal neurons have multiple subtypes (e.g. there are greater than 10 types of amacrine cells depending on the species), and not all of these subtypes express gap junctions. So it is unclear what this method would offer over conventional dye-injection methods, where tracers are introduced into a single cell. 

This is a fair criticism, the cut-loading technique is limited in that it cannot be used to determine the path of dye diffusion through coupled cells/networks, and is best suited to highly coupled networks. This limitation is highlighted in our discussion

Line 569: ‘A limitation of the cut-loading method, is that it cannot be used to delineate the path of dye-travel, as information detailing whether cells are homologously or heterologously coupled is not attainable.’

However, cut-loading is useful in that it enables the simultaneous assessment of coupling across all cells loaded across a large region of the retina. Retinae may then be counter labelled with antibodies specific to certain cell types (such as the nNOS expressing amacrine cells included in the present study) and the rate of coupling within these separate cell types may be compared in the same location and under the same conditions.’

Second, different neuronal subtypes in the retina are coupled to each other (e.g. amacrine cells and certain RGCs). Also, coupling exists between glial cells in the retina, and may even be unidirectional, as shown by Robinson et al (1993) and Newman and others (1997). How will the cut-loading method establish which of these cell types was first loaded? (On a side note, Cx43 is expressed in glial cells and not in neurons as stated in the Introduction)

As you have pointed out in your comments below, non-permeable dyes such as Rhodamine dextran may be added to the cut solution in order to determine which cells were loaded first. In line with your recommendations below we have conducted cut-loading on four additional retinae including 5% Rhodamine dextran to identify the cells initially loaded (see response below for more details). 

The prospect of unidirectional coupling is intriguing. It is possible that the cut-loading protocol may be modified to assess the directionality of gap-junctions by making two parallel cuts (with scalpel blades dipped in different molecular tracers), and measuring the diffusion of the separate dyes through the same cell network in both directions. However, for the purposes of the present study, we found it was sufficient to assess the diffusion of neurobiotin in large networks such as aHCs, as this presented the clearest and most obvious point of comparison for the three analytical techniques employed here. Additionally, coupling between aHCs is readily inhibited by drugs such as MFA, allowing for coupling comparisons to take place within a single incubation time. 

The paper would have improved considerably, if the authors chose to include a gap junction-impermeable large molecular weight dye (such as rhodamine dextran) or other gap junction permeable dyes (such as lucifer yellow). These latter studies will permit assessment of coupling coefficients using a positive and negatively charged molecules.

As indicated in your comment, the approach employed here only ascertains the rate of Neurobiotin dye transfer, which is determined by the connexin type, density and permeability. You are right in that cut-loading may be expanded by incorporating dyes of varying molecular weights to better investigate the permeability of the specific connexin types located between various cell types. As per your recommendation, we have repeated cut-loading at 1, 3, 5 and 10 minute incubation times with 5% Rhodamine dextran b m.wt 10000, to observe the cells initially loaded with dye and have incorporated example images in Fig 5f, g. This additional experiment allowed us to observe the fluorescence of aHCs filled first during cut-loading. Consistent with our previous assessment that the cut acts as a dye-sink and reduces the fluorescence of highly coupled aHCs adjacent to the cut with increasing incubation time, we observed little or no fluorescence in aHCs first filled by cut-loading at 3 and 10 mins. The fluorescence of neighbouring cells then decreases between 3 – 10 minutes as indicated by Fig 5f, g. 

Updated methods:

Line 162: ‘The retinas were briefly removed from solution and cut along the superior, temporal, and inferior axes with a size 11 scalpel blade that prior to each cut was dipped in 3% the biotin derivative, N-(2-aminoethyl) biotinamide hydrochloride (Neurobiotin™ Tracer, catalogue: SP-1120, Vector Laboratories, CA, USA) diluted in Ames solution. 5% wt/v Rhodamine dextran B 10,000 MW (catalogue: D1824, ThermoFisher Scientific, MA, USA) was added to the cut solution for one retina from each of the 1, 3, 5 and 10 minute incubation groups in experiment 1.’

Updated results: 

Line 384: ‘The fluorescence of cells situated immediately adjacent to the cut was less than that observed deeper into the tissue for all incubation conditions (Fig 5h), resulting in non-linear dye-transfer characteristics (compare open and closed markers in Fig 6a, b, c). The aHCs initially loaded during cut-loading are identified by the presence of the impermeable dye Rhodamine dextran (Fig 5f, g red fluorescence, aHCs identified by red arrows). Interestingly, these aHCs display little or no fluorescence from Neurobiotin following 3 and 10 minutes tissue incubation. Furthermore, the distance from the cut where reduced aHC fluorescence occurred, increased with longer incubation times (Fig 5h), suggesting that the cut in the aHC network acts as a dye-sink, allowing dye to be lost into the bath solution. To track true coupling coefficients, each equation was only fitted to the curve following the peak in cell fluorescence (closed markers Fig 6).’

---

## [Decision Letter · Decision Letter 1]

7 Jul 2022

Analytical methods for assessing retinal cell coupling using cut-loading

PONE-D-22-03044R1

Dear Dr. Myles,

We’re pleased to inform you that your manuscript has been judged scientifically suitable for publication and will be formally accepted for publication once it meets all outstanding technical requirements.

Kind regards,

Alexandre Hiroaki Kihara, Ph.D.

Academic Editor

PLOS ONE

Additional Editor Comments (optional):

Reviewers' comments:

Reviewer's Responses to Questions

**Comments to the Author**

1. If the authors have adequately addressed your comments raised in a previous round of review and you feel that this manuscript is now acceptable for publication, you may indicate that here to bypass the “Comments to the Author” section, enter your conflict of interest statement in the “Confidential to Editor” section, and submit your "Accept" recommendation.

Reviewer #1: All comments have been addressed

Reviewer #2: All comments have been addressed

Reviewer #3: All comments have been addressed

2. Is the manuscript technically sound, and do the data support the conclusions?

Reviewer #1: Yes

Reviewer #2: Yes

Reviewer #3: Yes

3. Has the statistical analysis been performed appropriately and rigorously? 

Reviewer #1: Yes

Reviewer #2: Yes

Reviewer #3: Yes

4. Have the authors made all data underlying the findings in their manuscript fully available?

Reviewer #1: Yes

Reviewer #2: Yes

Reviewer #3: Yes

5. Is the manuscript presented in an intelligible fashion and written in standard English?

Reviewer #1: Yes

Reviewer #2: Yes

Reviewer #3: Yes

6. Review Comments to the Author

Reviewer #1: (No Response)

Reviewer #2: (No Response)

Reviewer #3: My previous concerns were addressed. The authors have added a new figure showing labeling of rhodamine dextran and have addressed the limitations of the technique.

7. PLOS authors have the option to publish the peer review history of their article (what does this mean?). If published, this will include your full peer review and any attached files.

Reviewer #1: No

Reviewer #2: No

Reviewer #3: No

---

## [Editor Report · Acceptance letter]

11 Jul 2022

PONE-D-22-03044R1 

Analytical methods for assessing retinal cell coupling using cut-loading 

Dear Dr. Myles:

I'm pleased to inform you that your manuscript has been deemed suitable for publication in PLOS ONE. Congratulations! Your manuscript is now with our production department. 

Kind regards, 

on behalf of

Dr. Alexandre Hiroaki Kihara 

Academic Editor

PLOS ONE